# Structural basis for a nucleoporin exportin complex between RanBP2, SUMO1-RanGAP1, the E2 Ubc9, Crm1 and the Ran GTPase

Vladimir Baytshtok[1,4], Michael A. DiMattia[1,3,4] & Christopher D. Lima [1,2] ✉

The human nucleoporin RanBP2/Nup358 interacts with SUMO1-modified RanGAP1 and the SUMO E2 Ubc9 at the nuclear pore complex (NPC) to promote export and disassembly of exportin Crm1/Ran(GTP)/cargo complexes. In mitosis, RanBP2/SUMO1-RanGAP1/Ubc9 remains intact after NPC disassembly and is recruited to kinetochores and mitotic spindles by Crm1 where it contributes to mitotic progression. RanBP2 binds SUMO1-RanGAP1/Ubc9 via motifs that also catalyze SUMO E3 ligase activity. Here, we resolve cryo-EM structures of a RanBP2 C-terminal fragment in complex with Crm1, SUMO1-RanGAP1/Ubc9, and two molecules of Ran(GTP). These structures reveal several interactions with Crm1 including a nuclear export signal (NES) for RanGAP1, the deletion of which mislocalizes RanGAP1 and the Ran GTPase in cells. Our structural and biochemical results support models in which RanBP2 E3 ligase activity is dependent on Crm1, the RanGAP1 NES and Ran GTPase cycling.

The nuclear pore complex (NPC) coalesces nucleoporin subunits and their phenylalanine-glycine (FG) repeats to present a physical barrier between the nucleus and cytoplasm[1]. Most cargo larger than ~40 kDa can be escorted through the NPC by karyopherins that bind cargo via exposed nuclear export signals (NES) or nuclear localization signals (NLS) and traverse the NPC by interacting with the FG repeats of nucleoporins[2–5]. Directionality and cargo release depend on a Ran GTPase gradient with Ran(GTP) in the nucleus and Ran(GDP) in the cytoplasm[4,6]. The Ran gradient is maintained by the cytoplasmic Ran GTPase activating protein RanGAP1 that stimulates GTP hydrolysis to convert Ran(GTP) to Ran(GDP) and the nuclear guanine exchange factor (GEF) RCC1 that catalyzes GDP-GTP exchange to generate Ran(GTP)[7,8]. The nucleoporin RanBP2 (also known as Nup358) resides on the cytoplasmic side of the NPC where it coordinates many of these activities[9–12]. RanBP2 includes FG repeats that interact with karyopherins, Ran binding domains (RBDs) that disengage Ran(GTP) from importin and exportin-cargo complexes, and motifs that interact with SUMO-modified RanGAP1[13–19].

Ubiquitin (Ub) and ubiquitin-like (Ubl) proteins, such as SUMO regulate cellular processes through covalent attachment to proteins or other substrates through a cascade of activities catalyzed by E1 activating enzymes, E2 conjugating enzymes and E3 protein ligases[20–22]. RanGAP1 was the first SUMO substrate identified[14,15,23,24] but unlike most substrates, RanGAP1 can remain bound to the SUMO E2 Ubc9 even after SUMO conjugation[24–28]. Indeed, it is the SUMO-RanGAP1/Ubc9 complex that is recruited to NPCs via RanBP2's IR1-M-IR2 region, which contains tandem internal repeat (IR) motifs, each of which includes a SUMO interaction motif (SIM) that binds SUMO and other elements that bind Ubc9[14,27,29]. Additionally, both IR1 and IR2 are able to catalyze SUMO E3 ligase activity when expressed separately or together within IR1-M-IR2[28,30]. SUMO E3 ligases include the SP-RING family that uses a RING domain to interact with charged SUMO ~ E2 ("~" denotes a thioester bond) and a second family, as exemplified by RanBP2 and ZNF451, that bind SUMO ~ E2 through SIMs and elements that recognize the E2[30–33]. These E3 ligases stimulate E2 discharge by positioning the charged Ub/Ubl-E2 in a closed conformation primed for conjugation (reviewed in refs. 21,22)[34–36].

[1]Structural Biology Program, Sloan Kettering Institute, Memorial Sloan Kettering Cancer Center, 1275 York Ave, New York, NY, USA. [2]Howard Hughes Medical Institute, 1275 York Avenue, New York, NY, USA. [3]Present address: Schrödinger New York, 1540 Broadway, 24th Floor, New York, NY, USA. [4]These authors contributed equally: Vladimir Baytshtok, Michael A. DiMattia. ✉e-mail: limac@mskcc.org

Three SUMO proteins are widely expressed in human cells: SUMO1, SUMO2 and SUMO3, with SUMO2 and SUMO3 sharing ~95% sequence identity[32,37]. RanBP2 exhibits a binding preference for SUMO1 over SUMO2/3 resulting in a more stable complex with SUMO1-Ran-GAP1/Ubc9 relative to SUMO2/3-RanGAP1/Ubc9[27,38–40]. RanBP2 exhibits binding specificity for SUMO1, but cellular substrates of the RanBP2 E3 ligase are typically modified by SUMO2/3[41,42] suggesting that RanBP2 is a receptor for SUMO1-RanGAP1/Ubc9 while retaining E3 ligase activity with SUMO2/3. Several models have been put forward for how RanBP2 functions as an E3 ligase[17,40,43], but how its structure and interactions with SUMO1-RanGAP1/Ubc9, Crm1 and the Ran GTPase impact its E3 ligase activities remain unclear.

RanGAP1 exists in both unmodified and SUMO-modified forms and shuttles between nucleus and cytoplasm, and while Crm1 is implicated as its exporter, an NES within RanGAP1 has not been identified[14,44,45]. Further, RanBP2 maintains interactions with SUMO1-RanGAP1/Ubc9[43,46] in mitosis and is recruited to kinetochores and spindles by Crm1[47], interactions that are perturbed by leptomycin B, a molecule that disrupts Crm1-NES interactions[47,48]. Here, we address a basis for these interactions through structural, cellular, and biochemical studies to reconstitute and characterize activities of complexes formed between RanBP2, SUMO1-RanGAP1, Ubc9, Crm1 and the Ran GTPase.

## Results

### Reconstitution and cryo-EM analysis of RanBP2/SUMO1-Ran-GAP1/Ubc9/Crm1/Ran(GTP)

Reconstitution of RanBP2/SUMO1-RanGAP1/Ubc9/Crm1/Ran complexes utilized a RanBP2 fragment first characterized in detail by Melchior and colleagues[17,43] that spans residues 2300–3060 and includes the IR1-M-IR2 motifs flanked by two FG repeats and two RBDs, the N-terminal RBD3 and C-terminal RBD4 (RanBP2$^{RBD3-4}$, Fig. 1A). The GTPase-defective variant Ran$^{Q69L}$(GTP), reported to stabilize interactions between Crm1 and RanBP2[17], co-migrated with RanBP2$^{RBD3-4}$/SUMO1-RanGAP1/Ubc9/Crm1 in our reconstitution but size exclusion

chromatography (SEC) revealed a broad peak consistent with poly-dispersity or conformational heterogeneity, perhaps due to multiple Ran binding sites presented by RanGAP1, the two RBDs, and potentially Crm1 (Fig. 1B, C, Supplementary Fig. 1). To determine if the two RBDs contributed to this behavior, truncations of RanBP2$^{RBD3-4}$ were generated: RanBP2$^{2300-2910}$ containing only RBD3 and RanBP2$^{2446-3060}$ containing only RBD4 (RanBP2$^{RBD3}$ and RanBP2$^{RBD4}$, respectively; Fig. 1A, B). RanBP2$^{RBD3}$ exhibited a broad elution profile similar to RanBP2$^{RBD3-4}$, but RanBP2$^{RBD4}$ eluted later than RanBP2$^{RBD3}$ and RanBP2$^{RBD3-4}$, with a 100 kDa reduction in apparent molecular weight, much larger than expected for deletion of a 17 kDa RBD domain (Fig. 1C, Supplementary Fig. 1). Additionally, as both RanBP2$^{RBD3}$ and RanBP2$^{RBD4}$ fragments are ~70 kDa and contain a single RBD, complexes containing these variants would be expected to have similar Stokes radii and SEC elution profiles, further suggesting that the different elution profile observed for the RanBP2$^{RBD4}$ complex may be a result of a more compact architecture or less conformational heterogeneity. Based on this, a RanBP2$^{RBD4}$/SUMO1-RanGAP1/Ubc9 complex was purified, incubated with excess Crm1 and Ran$^{Q69L}$(GTP), and purified again by gel filtration (Supplementary Fig. 2A, B). Prior to vitrification, the sample was mildly crosslinked using bis(sulfosuccinimidyl)suberate (BS3) to stabilize complexes (Supplementary Fig. 2C–E).

Cryo-EM data collection and processing produced ~700 k unique particles containing Crm1 (Supplementary Figs. 2F–M, 3, and Supplementary Table 1). Crm1/Ran$^{Q69L}$(GTP) was separated from unbound Crm1, resulting in ~535 k particles and a reconstruction with an overall resolution of 3.18 Å with densities evident for RanBP2, portions of RanGAP1 and other diffuse densities emanating from the Crm1 ring (Fig. 2A and Supplementary Figs. 2, 3 – Overall map I). Focused refinement of Crm1/Ran$^{Q69L}$(GTP) improved the overall resolution to 2.89 Å. 3D classification revealed that the RanGAP1 GAP domain was highly mobile, anchored to Crm1 via an N-terminal helix, and bound to Ran$^{Q69L}$(GTP) and the RanBP2 RBD4. Further processing resulted in reconstructions using ~290 k particles for the RDB4/

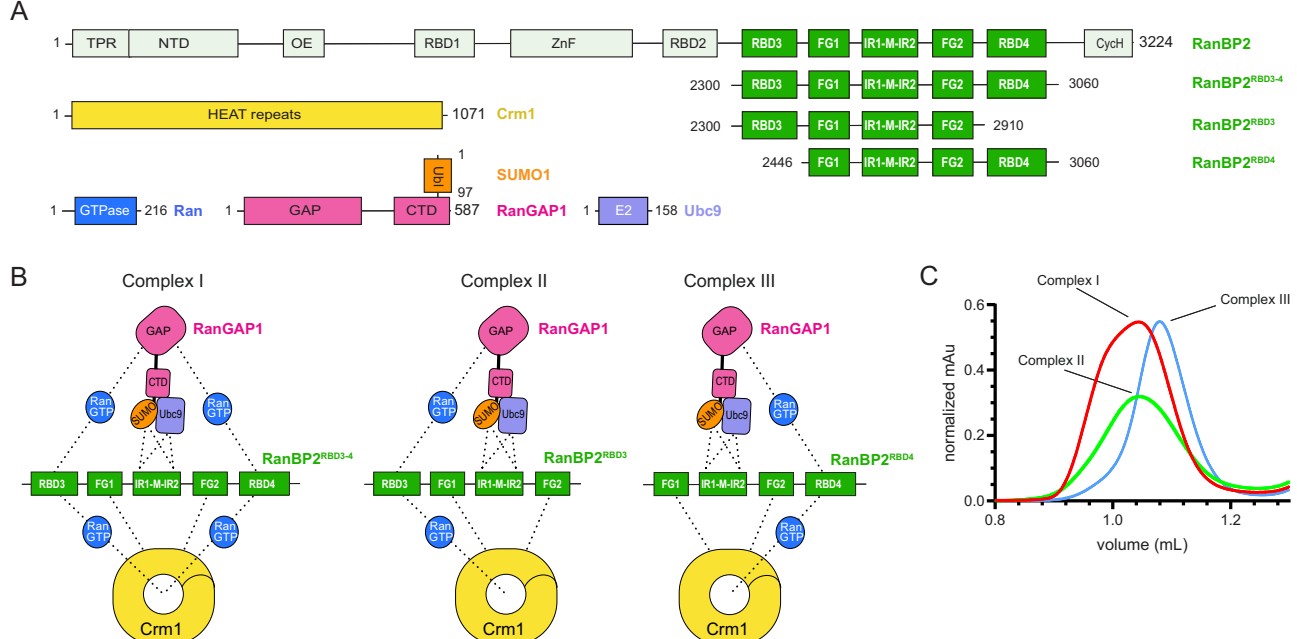

**Fig. 1 | Reconstitution of RanBP2/SUMO1-RanGAP1/Ubc9/Crm1/Ran$^{Q69L}$(GTP) complexes. A** Components and domain organization: full length RanBP2 and fragments, Crm1, Ran GTPase, RanGAP1 conjugated to SUMO1, and the E2 Ubc9. RBD = Ran binding domain; IR = internal repeat; FG = phenylalanine-glycine repeats. **B** Schematic indicating multivalent protein-protein interactions in complexes assembled with RanBP2 variants containing RBD3 and RBD4 (Complex I), RBD3 (Complex II), or RBD4 (Complex III). Dashed lines indicate possible interactions. **C** Profiles obtained by size exclusion chromatography for complexes I–III shown in B. See Supplementary Fig. 1 for full SEC profiles and SDS-PAGE gels of individual components and assembled complexes.

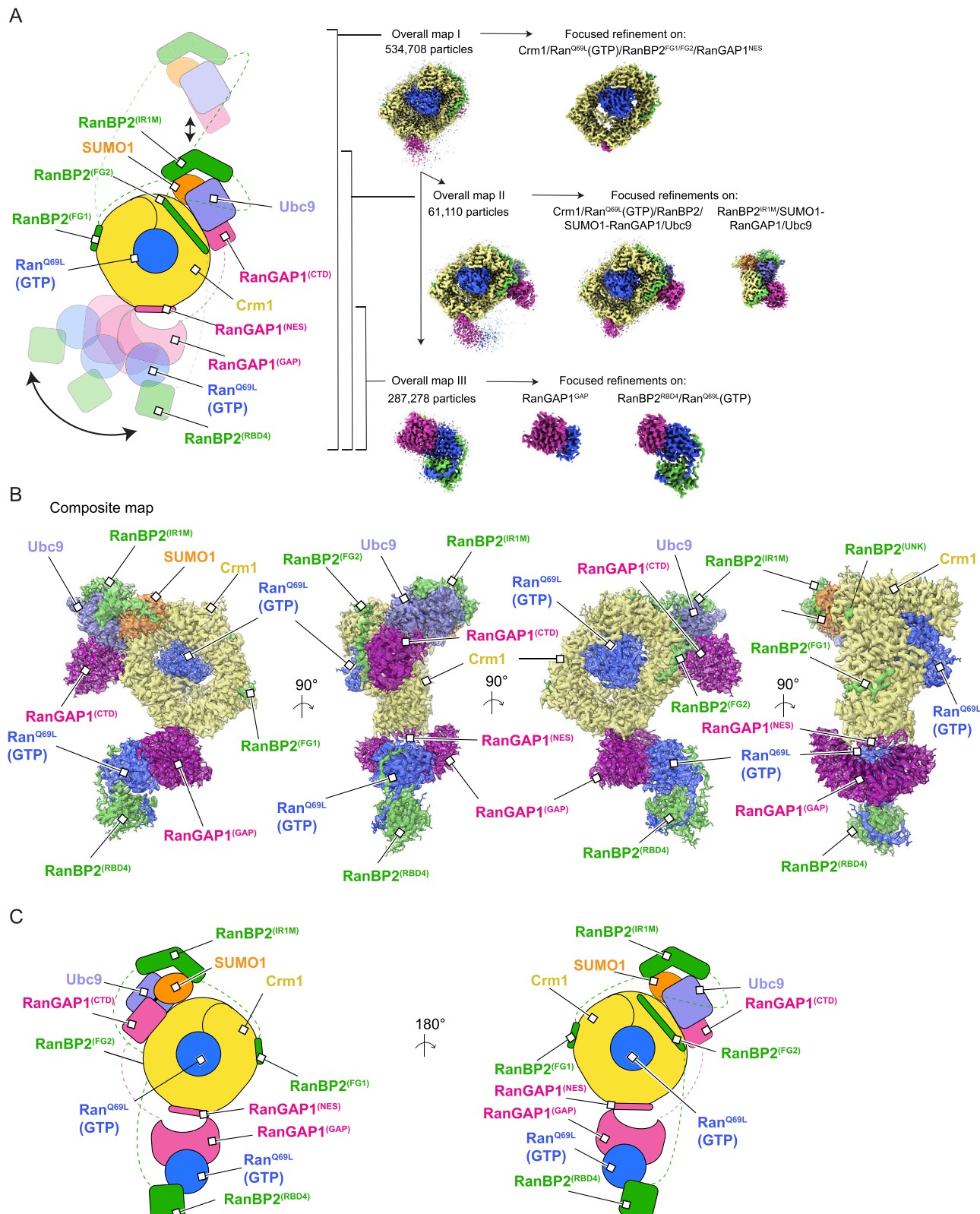

Ran^Q69L(GTP)/RanGAP1^GAP subcomplex at an overall resolution of 3.52 Å (Fig. 2A, Supplementary Figs. 2, 3 – Overall map III), with focused refinements of RBD4/Ran^Q69L(GTP) at 3.52 Å and the GAP domain at 3.37 Å. The C-terminal domain (CTD) of RanGAP1, SUMO1 and Ubc9 were resolved after signal subtraction and 3D classification of 74 k particles to generate reconstructions from 61 k particles for the RanBP2/Ubc9/SUMO1-RanGAP1/Crm1/

Ran^Q69L(GTP) subcomplex at an overall resolution of 3.40 Å (Fig. 2A, Supplementary Figs. 2, 3 - Overall map II) with focused refinements of RanBP2/Ubc9/SUMO1-RanGAP1^CTD and Crm1/Ran^Q69L(GTP) at 3.29 and 3.10 Å, respectively. As components of the complex exist in multiple conformations (Fig. 2A, Supplementary Figs. 2, 3), these reconstructions were combined to generate a composite map for model building and refinement.

**Fig. 2 | Cryo-EM structures of RanBP2$^{RBD4}$/SUMO1-RanGAP1/Ubc9/Crm1/Ran$^{Q69L}$(GTP). A** Cartoon representation of the RanBP2$^{RBD4}$/SUMO1-RanGAP1/Ubc9/Crm1/Ran$^{Q69L}$(GTP) complex illustrating conformational flexibility of individual components (left). The RanGAP1$^{GAP}$/Ran$^{Q69L}$(GTP)/RanBP2$^{RBD4}$ subcomplex rotates with respect to Crm1/Ran$^{Q69L}$(GTP) around the fixed RanGAP1$^{NES}$ anchor (bottom double-headed arrow) while the RanBP2$^{IR1}$/Ubc9/SUMO1-RanGAP1$^{CTD}$ subcomplex is bound to Crm1/Ran$^{Q69L}$(GTP) in only a fraction of the observed particles (top double-headed arrow). Bracketed regions of the cartoon point to the corresponding density maps with number of particles used to reconstruct these maps indicated. Processing of overall map I was used to generate overall map II and, separately, overall map III, as indicated by the arrows. Focused refinements of individual subcomplexes were derived from the overall maps and are indicated to their right. See Methods and Supplementary Figs. 2, 3 for more details. Individual components colored and labeled as indicated. Dashed lines indicate disordered regions. **B** Orthogonal views of the composite map and models. EM density (contoured at level 9) and atomic model overlaid and components individually colored and labeled as indicated. RanBP2$^{(UNK)}$ indicates unassigned density likely belonging to RanBP2 as discussed in the text. **C** Cartoon representation of the composite map colored as in (**A** and **B**).

## Overall architecture

The architecture of the RanBP2$^{RBD4}$/SUMO1-RanGAP1/Ubc9/Crm1/2xRan$^{Q69L}$(GTP) complex reveals Crm1 and RanBP2 as major organizing factors of the complex (Fig. 2B, C). At the center is Crm1 bound to one molecule of Ran$^{Q69L}$(GTP) with SUMO1-RanGAP1$^{CTD}$/Ubc9 and RanGAP1$^{GAP}$/Ran$^{Q69L}$(GTP)/RanBP2 RBD4 subcomplexes on the periphery of the Crm1 ring. Crm1 adopts a closed toroidal, Ran(GTP) and cargo-bound conformation similar to that described previously (PDB 3GJX, 3NBY and 5DIS)[49–52]. The RanBP2 FG1-IR1-M-IR2-FG2-RBD4 fragment wraps around the complex with the two FG domains, defined as FG1 and FG2, positioned on opposite sides of the Crm1 ring. The RanBP2 IR1 motif is bound to SUMO1-RanGAP1/Ubc9 and flanked by disordered segments that link it to FG1 and FG2. Following FG2 is another disordered segment that separates FG2 from RBD4. The latter is bound to a molecule of Ran$^{Q69L}$(GTP) that is in turn bound by the GAP domain of RanGAP1. RanGAP1 includes an N-terminal leucine-rich repeat GAP domain and a CTD that are separated by more than 50 Å and connected by a ~70 residue acidic linker that is not observed in reconstructions.

## Crm1 interacts with Ran$^{Q69L}$(GTP) and the FG repeats of RanBP2

Interactions between Crm1 and Ran$^{Q69L}$(GTP) in our complex resemble those previously reported and will not be discussed further[51,53]. FG regions of RanBP2$^{RBD4}$ were reported to contribute to Crm1 binding[17] and continuous densities for FG1 residues 2507–2517 and FG2 residues 2841–2881 are observed, respectively (Fig. 3A, B, Supplementary Fig. 4 panels 6 and 7). Residues from FG1 interact with outer helices (A helices) of HEAT (Huntingtin, EF-3, PP2A, TOR1) repeats 14–16 of Crm1, burying ~1280 Å$^2$ of combined surface area with several hydrophobic FG1 residues acting as major contributors to the interface (Fig. 3A). RanBP2 FG2 includes six phenylalanines and one tryptophan within residues 2841–2881 that engage Crm1 HEAT repeats 2–7, burying a total surface area of ~3630 Å$^2$ (Fig. 3B–D).

Structures of other proteins containing FG repeats have been determined in complex with Crm1[51,53,54]. FG repeats from human Nup214 (PDB 5DIS)[51] and yeast Nup42p (PDB 5XOJ)[53] also bind within pockets created by Crm1 helices 14–16, although they adopt distinct conformations or polarities relative to each other or to RanBP2 FG1 (Supplementary Fig. 5A). Of note, Phe114 of Nup42p and Phe1922 of Nup214 dock into the same hydrophobic groove of Crm1 as Phe2514 of RanBP2 FG1 (Supplementary Fig. 5A). FG repeats from Nup214 continue to wrap around Crm1, binding in several hydrophobic pockets formed by Crm1 HEAT repeats 17–20, some of which are also bound by FG repeats from Nup42p and Yrb2p (yeast RanBP3, PDB 3WYF)[54]. Continuous densities are not observed for RanBP2 past Gly2517, but isolated densities consistent with phenylalanine side chains are observed in pockets formed by HEAT repeats 18 and 19 where Nup214 Phe1947 binds Crm1 (Supplementary Fig. 5B left panel) and HEAT repeats 17 and 18 where Phe1982 of Nup214, Phe92 of Nup42p, and Phe100 of Yrb2p bind Crm1 (Supplementary Fig. 5B right panel). Interactions between Crm1 and FG repeats of Nup214 and Yrb2p overlap with RanBP2 FG2 residues 2841-2853 as illustrated by comparing Phe2842 and Phe2850 of RanBP2 with phenylalanine residues of Nup214 and Yrb2p (Supplementary Fig. 5C). FG2 in our structure, however, continues to wrap around Crm1 HEAT repeats 5-7, a feature not observed in other structures for complexes between FG regions and Crm1. While the SUMO1-RanGAP1/Ubc9 subcomplex is resolved in only a subset of our particles, FG2 residues Lys2868-Thr2872 are observed in all reconstructions and are within 7-8 Å of the C-terminal helix of RanGAP1.

## RanGAP1 forms a complex with Ran$^{Q69L}$(GTP) and RBD4 of RanBP2

The GAP domain of RanGAP1 is bound to a molecule of Ran$^{Q69L}$(GTP) that in turn interacts with the C-terminal RBD4 of RanBP2 (Fig. 4A). This subcomplex structure is similar to the only currently available structure of RanGAP1 in complex with Ran, a crystal structure of the *S. pombe* RanGAP1 in complex with human Ran(GMPPNP) and the RBD of RanBP1 (PDB 1K5D)[55]. While similar, notable differences include the absence of the N-terminal human RanGAP1 helix which is not conserved in fission yeast, the inactivating Q69L mutation in Ran, and subtle differences in the way RBD4 engages Ran relative to the RBD of RanBP1 which are likely due to differences in primary sequences (~55% seq identity). Additionally, Arg2975 and Lys2980 of RanBP2 RBD4 penetrate deeper into a Ran interface where their side chain nitrogen atoms contact the carbonyl oxygen atoms of Arg29 and His30 (Fig. 4B), part of the switch I region of Ran, which undergoes large conformational changes in response to nucleotide binding and identity. Another notable difference is in the RanGAP1-Ran interface where the ε-amine of Arg93 of human RanGAP1, in addition to forming a salt bridge with Glu46 as also observed in the *S. pombe* structure, contacts Ala41 of Ran, which is within 4-5 Å from the γ-phosphate of GTP and the coordinating Mg$^{2+}$ ion (Fig. 4C). Additional hydrogen bonding interactions present in our complex but not observed in the *S. pombe* RanGAP1-Ran complex involve Asp273 of RanGAP1, which forms a hydrogen bonding network with Ser135 and Arg95 of Ran (Fig. 4D).

Based on the *S. pombe* RanGAP1-Ran(GMPPNP)-RanBP1 structure, it was proposed that in the absence of a classical catalytic Arg finger, RanGAP1 activates Ran for GTP hydrolysis by stabilizing the switch II loop of Ran, thereby properly orienting the catalytic glutamine residue within Ran (Gln69 in human Ran)[55,56]. Our structure confirms this model and expands upon it via comparison to the currently available structures for human Ran(GMPPNP) bound to RBD1, RBD2, RBD3, and RBD4 domains of RanBP2 in the absence of RanGAP1[11,57]. Inspection of the Ran switch II loop in these structures shows that it adopts conformations that would clash with RanGAP1 in our human complex and in *S. pombe* structures (Fig. 4E–G). Thus, as has been reported previously, RanGAP1 stabilizes the switch II loop of Ran[55,56] but we further note that it does so by restricting conformational flexibility of the switch II loop, which in turn orients the catalytic glutamine and neighboring switch II residues to catalyze GTP hydrolysis in the absence of a catalytic Arg finger.

## RanGAP1 contacts Crm1 via two distinct interfaces

The N-terminal RanGAP1 GAP domain is tethered to Crm1 by its N-terminal helix, which occupies the hydrophobic Crm1 NES-binding pocket between the A helices from HEAT repeats 11 and 12 (Fig. 5A–D, Supplementary Fig. 4 panel 10) and overlaps with Crm1-bound NES

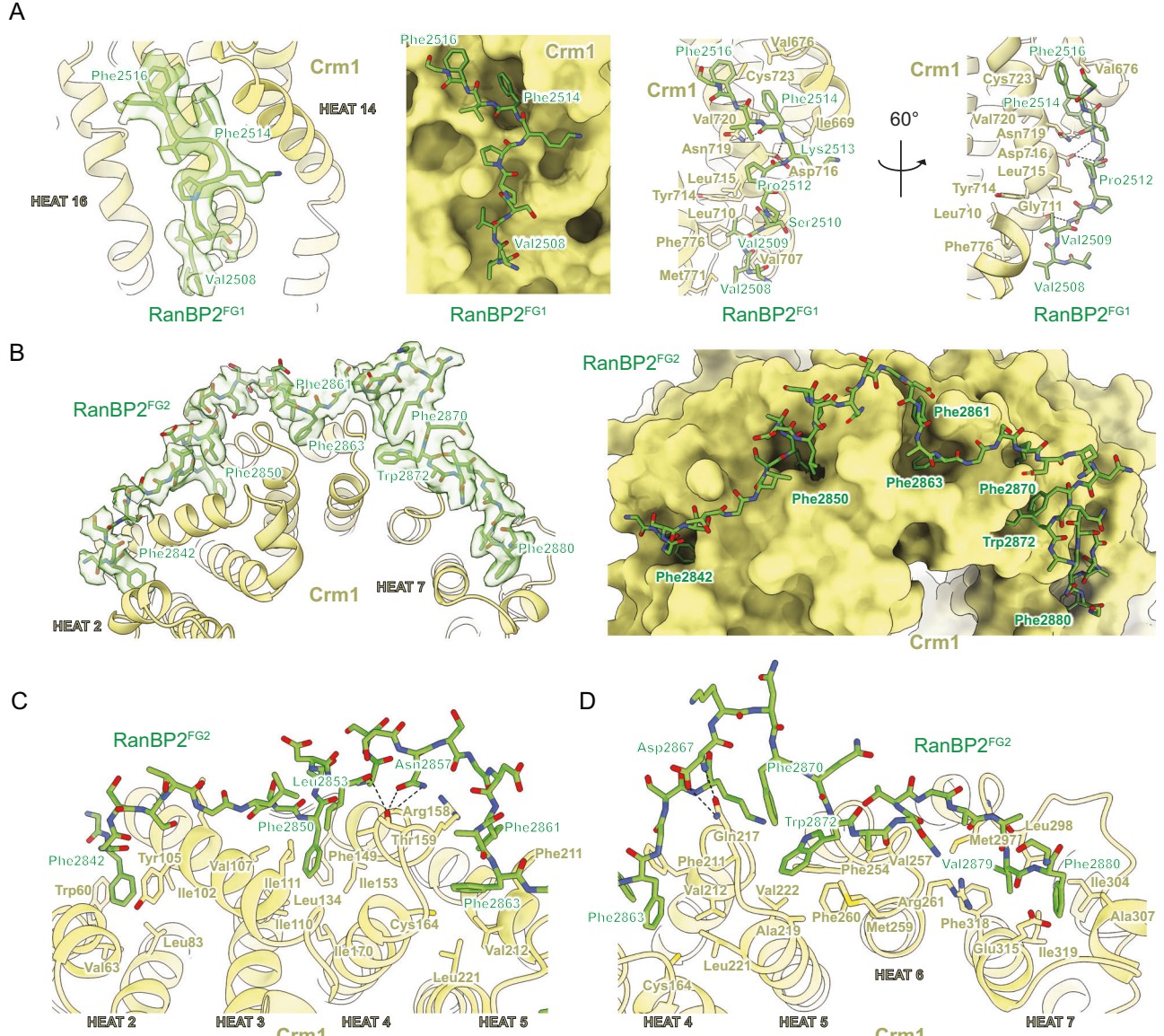

**Fig. 3 | FG1 and FG2 repeats of RanBP2 bound to Crm1. A** EM density for RanBP2 FG1 (contoured at level 6) overlaid with model to highlight interactions with Crm1 HEAT repeats 14–16, depicted in cartoon representation (left panel); FG1 residues in stick representation and Crm1 as surface representation to emphasize deep pockets on its surface (second panel); the FG1-Crm1 interface with side chains and potential hydrogen bonds shown (dashed lines, third and fourth panels). **B** EM density (contoured at level 5.5) and RanBP2 FG2 residues highlighting interactions with Crm1 HEAT repeats 2–7, depicted in cartoon representation (left panel). Similar view of FG2 model with Crm1 depicted as surface representation to emphasize deep pockets on its surface (right panel). **C** FG2 (stick representation) interactions with Crm1 HEAT repeats 2–5 cartoon and stick representation for side chains) with potential hydrogen bonds depicted as dashed lines. **D** Detailed view of the interface between FG2 and Crm1 HEAT repeats 4–7 depicted as in panel C.

motifs from protein kinase inhibitor (PKI) and Snurportin1 (SPN1; Supplementary Fig. 5D)[51,53]. The RanGAP1 N-terminal helix is conserved in vertebrate and invertebrate species, although less strongly in insects, and is not conserved in organisms such as *S. cerevisiae*, *S. pombe* and *A. thaliana* (Fig. 5E).

Unlike the N-terminal GAP domain, which is observed in most particles, the CTD of RanGAP1 is observed in only ~14% of particles suggesting high mobility within this complex (Fig. 2A and Supplementary Fig. 3). Despite this mobility, reconstructions show that the C-terminal RanGAP1 domain is conjugated to SUMO1 via Lys524 and bound to Ubc9 with Ubc9 and SUMO1 coordinated by the RanBP2 IR1 module (Supplementary Fig. 4 panels 2 and 3). The structure of the SUMO1-RanGAP1/Ubc9/ RanBP2 IR1 subcomplex shares overall similarities to a crystal structure of human SUMO1-RanGAP1^CTD/Ubc9/RanBP2^IR1-M (PDB

1Z5S, RMSD = 0.897)[27]. In addition, a new interface is observed between Crm1 and the SUMO1-RanGAP1^CTD/Ubc9/RanBP2^IR1 subcomplex consisting of the two C-terminal RanGAP1 helices (residues 557–585) and Crm1 HEAT repeats 5–7, with additional contributions from Ubc9, interactions that bury approximately 1960 Å² of combined surface area (not including the RanGAP1-Ubc9 interface; Fig. 5F–H). This interface does not appear to contribute to specific recognition of SUMO1 as no contacts are observed between Crm1 and SUMO1.

As contacts between Crm1 and the N-terminal helix of the Ran-GAP1 GAP domain and the SUMO1-RanGAP1/Ubc9 subcomplex were unexpected, we sought to confirm if any of these interactions could be recapitulated by AlphaFold multimer (version 2.2), which was released after our structures were determined[58]. Indeed, even in the absence of RanBP2 and Ran(GTP), AlphaFold predicts interactions between Crm1

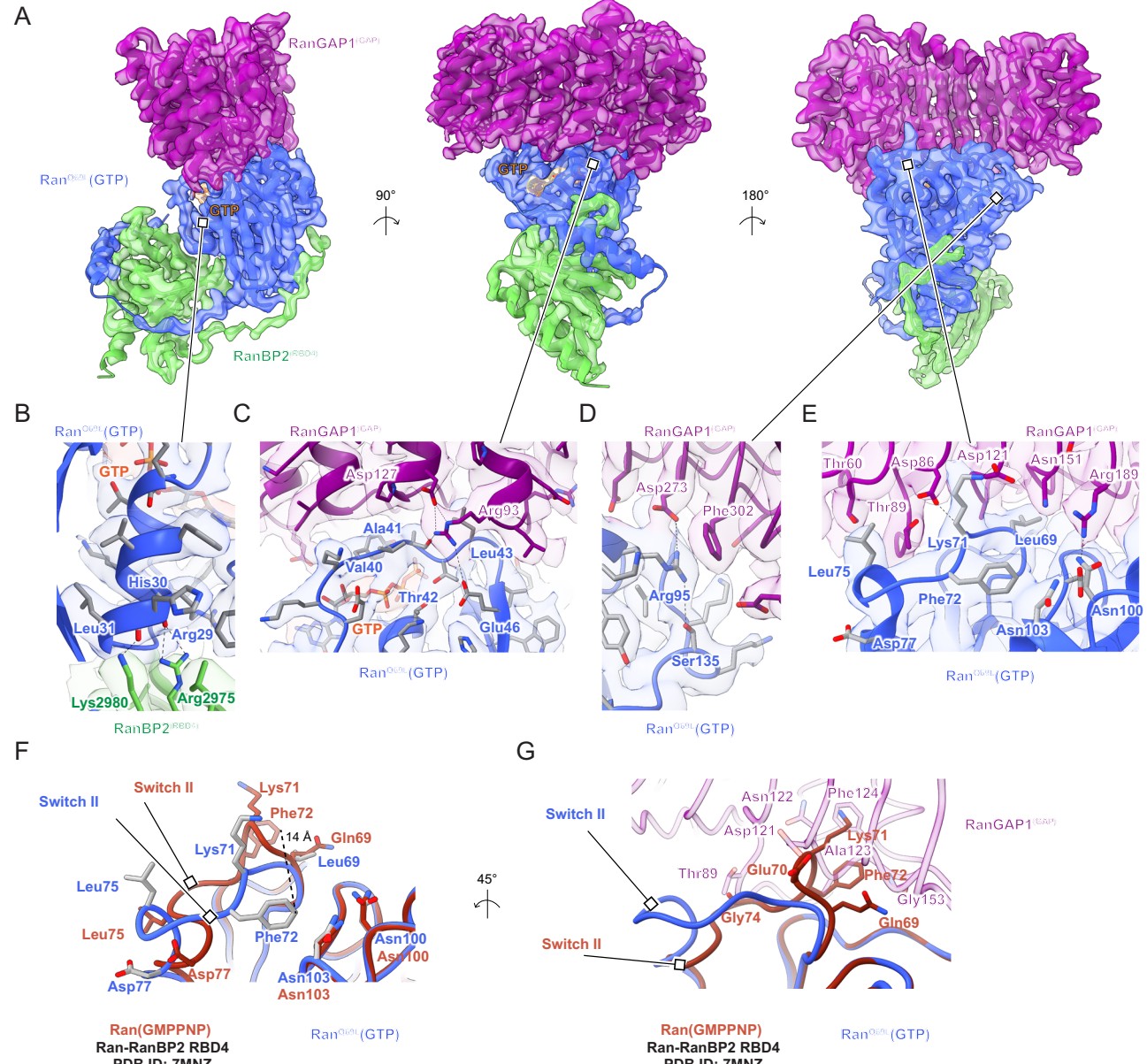

**Fig. 4 | RanGAP1$^{GAP}$/Ran$^{Q69L}$(GTP)/RanBP2$^{RBD4}$ subcomplex structure.**
**A** Orthogonal views with EM density (contoured at level 9) overlaid with cartoon representation. Proteins are labeled and colored as in Fig. 2, GTP EM density colored red, lines pointing to magnified views in (**B–E**). **B** Magnified view of interactions between RanBP2$^{RBD4}$ Arg2975 and Lys2980 and the switch I helix of Ran$^{Q69L}$(GTP). EM density (contoured at level 9.2) overlaid on cartoon model with residues in stick representation and potential hydrogen bonds indicated as dashed lines. **C** Magnified view of interactions of Arg93 of the GAP domain and its proximity to the γ-phosphate of GTP and Ala41 and Glu46 of the switch I region of Ran$^{Q69L}$(GTP). EM density (contoured at level 8), model representation as in (**B**). **D** Magnified view of interactions between Ran$^{Q69L}$ Ser135, Arg95 and RanGAP1

Asp273, Phe302. EM density (contoured at level 8), model representation as in (**B**). **E** Magnified view of the switch II loop of Ran$^{Q69L}$(GTP) and the RanGAP1 GAP domain. EM density (contoured at level 9.2), model representation as in (**B**). **F** Overlay of Ran models from the Ran(GMPPNP)-RanBP2$^{RBD4}$ complex (dark orange; PDB 7MNZ)[11] and Ran$^{Q69L}$(GTP) from our RanGAP1$^{GAP}$-Ran$^{Q69L}$(GTP)-RanBP2$^{RBD4}$ complex (coloring as in A-E) to highlight conformational differences in switch II loops, including the 14 Å movement of Phe72 (dashed line; same view as in (**E**)). **G** Rotation of the view in (**F**) with part of the RanGAP1 interface from our structure included to highlight potential clashes between the indicated residues from RanGAP1 and from the switch II loop of Ran(GMPPNP) from 7MNZ. Residues from Ran$^{Q69L}$(GTP) from our structure omitted for clarity.

and the N-terminal helix of the GAP domain in 17 of 25 predictions (Supplementary Fig. 5E). Perhaps more surprising given that it was only present in 14% of our particles, it also predicts a similar interface between Crm1 and the CTD of RanGAP1 in 24 of 25 predictions (Supplementary Fig. 5E).

### The N-terminal helix of RanGAP1 is a nuclear export signal

Prior studies in yeast and human cells showed that RanGAP1 shuttles between the nucleus and cytoplasm[14,44,45,59]. In addition, localization

of human RanGAP1 is perturbed by leptomycin B (LMB) suggesting a dependency on Crm1 for export[44]. Our structures reveal that the N-terminal helix of RanGAP1 occupies the hydrophobic Crm1 NES-binding pocket (Fig. 5B–D, Supplementary Fig. 4 panel 10)[51,53], suggesting that it may function as an NES for human RanGAP1. The RanGAP1 N-terminal helix projects Ile6, Leu9, Leu13, Thr16, and Val18 into Crm1, which can be aligned to the x-φ$^0$-x$_2$-φ$^1$-x$_3$-φ$^2$-x$_2$-φ$^3$-x-φ$^4$ NES motif described for PKI-type NES signals, where x is any residue and φ is hydrophobic (Fig. 5B–D)[50]. One exception is Thr16 at φ$^3$, a

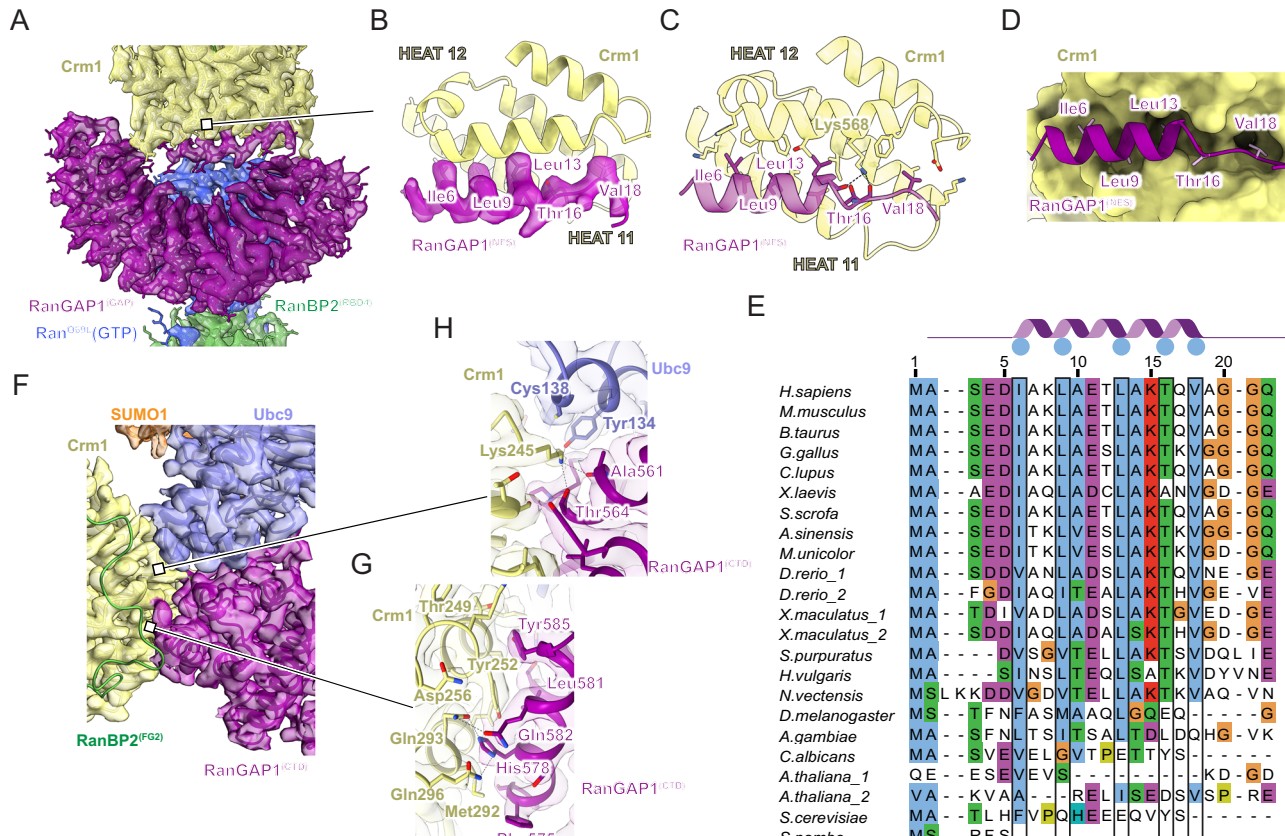

**Fig. 5 | RanGAP1 interacts with Crm1 via its GAP and CTD domains. A** The N-terminal helix of RanGAP1 interacts with Crm1. Coloring, EM density (contoured at level 9) and model representation as in Fig. 2B (rightmost panel). Line points to magnified view in (**B**). **B** Magnified view of (**A**) to highlight EM density (contoured at level 8) for key residues of the RanGAP1 NES that dock into the Crm1 pocket formed by helices from HEAT repeats 11 and 12. **C** Same view as in (**B**) but showing side chain contacts and potential hydrogen bonds between the RanGAP1(NES) and Crm1 surface. **D** Similar view as in (**C**) depicting the RanGAP1(NES) in cartoon representation and a surface representation of Crm1 to highlight the NES binding pocket. **E** Multiple sequence alignment of RanGAP1 N-termini across 20 indicated species. Blue circles in cartoon above the alignment and boxes indicate position of residues within the NES motif. Numbering shown for *H.* *sapiens* RanGAP1. **F** Magnified views of the Crm1/RanGAP1(CTD)/Ubc9 interface shown in the same orientation as in Fig. 2B (third panel from the left). EM density (contoured at level 10) overlaid with cartoon representation for Crm1, Ubc9, RanGAP1(CTD) and SUMO1. EM densities omitted for RanBP2(FG2) to enable visualization of the interface. Lines pointing to magnified view in (**G**) and (**H**). **G** Magnified view of (**F**) highlighting interactions between the C-terminal helices of RanGAP1(CTD) and Crm1. EM density (contoured to level 14) and model representations as in (**C**) with potential hydrogen bonds as dashed lines. **H** Magnified view of (**F**) highlighting Crm1 interactions and Lys245 in the interface between RanGAP1(CTD) and Ubc9. EM density (contoured to level 12) with models represented as in (**G**).

position typically occupied by Phe or Leu in prototypical NES motifs. In addition, hydrogen bonding interactions are evident between the Crm1 Lys568 ε-amino group and the backbone carbonyls of Ala14 and Thr16 of RanGAP1 (Fig. 5C).

To determine if the N-terminal 18 amino acids of RanGAP1 are sufficient for nuclear export, human hTERT-RPE1 cell lines were generated that stably express monomeric EGFP fused to wild type N-terminal residues of RanGAP1 or a mutant sequence where four of the key NES motif hydrophobic residues (Ile6, Leu9, Leu13, and Thr16) are replaced by alanine (4 A mutant) (Fig. 6A). While RanGAP1(1-18-WT)-mEGFP was predominantly cytoplasmic, RanGAP1(1-18-4AMUT) was observed in the nucleus and cytoplasm, suggesting that the N-terminal residues of RanGAP1 can induce nuclear export when fused to EGFP and that residues Ile6, Leu9, Leu13, and Thr16 of RanGAP1 contribute to export (Fig. 6A).

To assess if the N-terminal helix of endogenous RanGAP1 functions as an NES in cells, we deleted residues 5-17 of RanGAP1 in hTERT-RPE1 cells via CRISPR-Cas9 (Fig. 6B). Isolated clones homozygous for the Δ5-17 deletion express a RanGAP1 variant where 4 of the 5 key hydrophobic residues of the putative RanGAP1 NES motif are deleted (ΔNES; Fig. 6C). Western blot analysis of lysates obtained

from these clones showed similar basal levels of unmodified and SUMO1-modified RanGAP1 in two of the three isolated clones and that RanGAP1 in those clones is phosphorylated after nocodazole arrest[60], suggesting that removal of the NES does not perturb known post-translational modifications of RanGAP1 (Fig. 6C, D). Characterization of clone 3 was not pursued further due to the lower levels of RanGAP1 observed in this clone (Fig. 6C). Confocal microscopy of fixed cells using an α-RanGAP1 antibody confirmed that cells expressing RanGAP1ΔNES exhibit staining of the nuclear rim similar to cells with wild type RanGAP1 (Fig. 6E). However, quantification of nuclear to cytoplasmic ratios (N/C) for RanGAP1 in wild type RPE1 and RanGAP1ΔNES cells revealed that RanGAP1ΔNES exhibits higher N/C ratios compared to wild type RanGAP1 suggesting an export defect for RanGAP1ΔNES (Fig. 6F). Treatment of cells with LMB resulted in higher N/C ratios for RanGAP1 in wild type RPE1 cells consistent with Crm1 mediating nuclear export of RanGAP1 and similar to the basal N/C ratios seen in RanGAP1ΔNES cells. In contrast, LMB treatment of cells expressing RanGAP1ΔNES resulted in a comparatively smaller increase in N/C ratios. These results are consistent with the N-terminal helix of RanGAP1 functioning as a Crm1-dependent NES in cells.

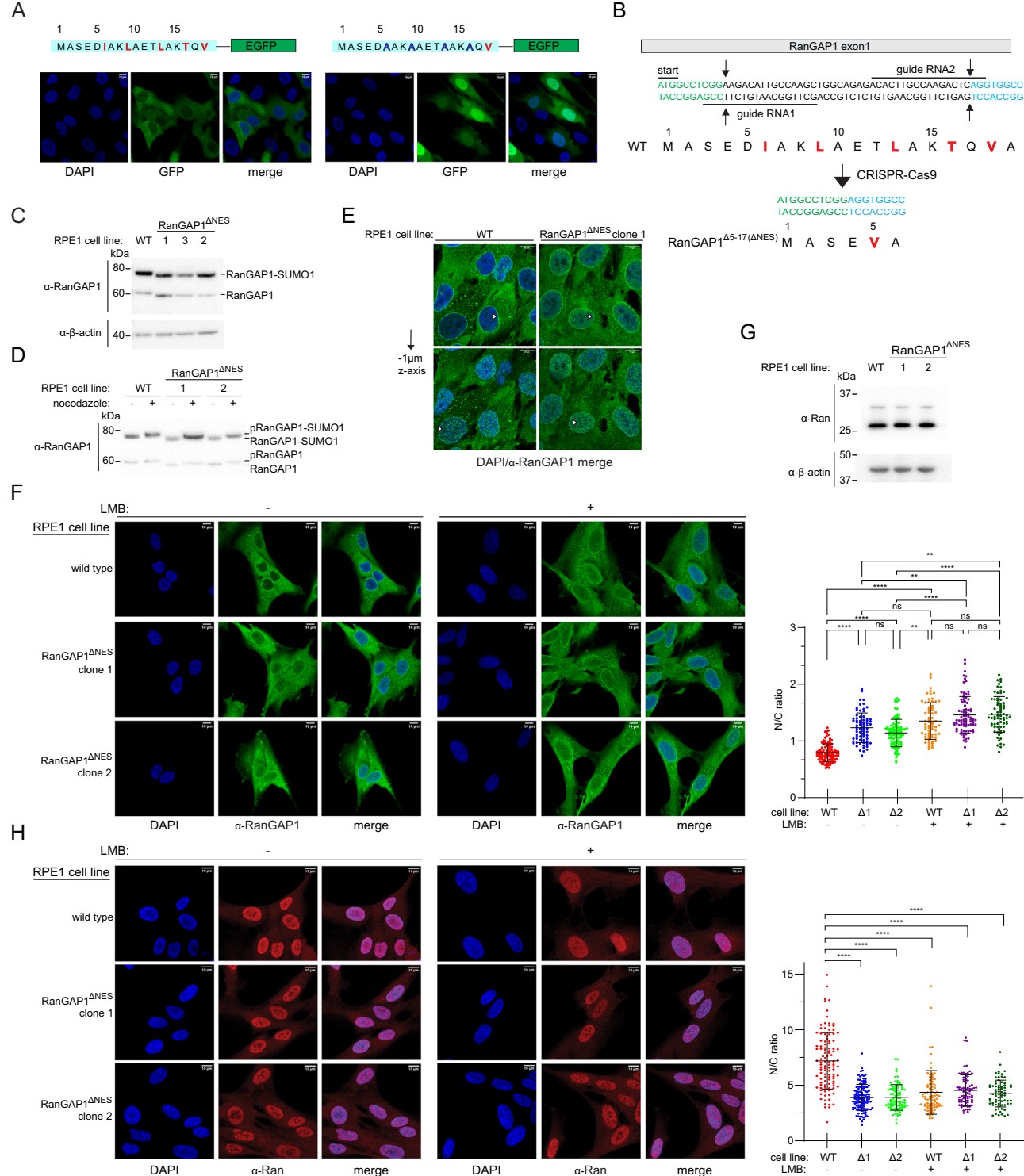

## Nuclear localization of Ran in cells expressing RanGAP1^ΔNES

Cells were next analyzed for localization of the Ran GTPase because RanGAP1 is the primary activator of the Ran GTPase in human cells and because RanGAP1^ΔNES is retained in the nucleus. While total levels of Ran remained unaffected (Fig. 6G), immunostaining using a Ran-specific antibody showed that Ran is redistributed from the nucleus to the cytoplasm in cells expressing RanGAP1^ΔNES compared to cells expressing wild type RanGAP1 (Fig. 6H, compare N/C ratios for WT vs Δ1 vs Δ2 in the absence of LMB). Treating wild type cells with LMB redistributed Ran from the nucleus to the cytoplasm, shifting N/C ratios close to those observed for cells expressing RanGAP1^ΔNES in the absence of LMB (Fig. 6H). By comparison, treating cells expressing RanGAP1^ΔNES with leptomycin B altered N/C ratios only slightly (Fig. 6H). These results suggest that nuclear retention of RanGAP1^ΔNES may contribute to mislocalization of Ran in these cells.

A Ran GTP-GDP gradient, where GTP-bound Ran is nuclear and GDP-bound Ran is cytoplasmic, is required for normal active nucleo-cytoplasmic transport (NCT)[6]. While immunofluorescence experiments conducted here cannot distinguish between GTP and GDP-bound Ran, we posited that mislocalization of Ran might result in perturbations in NCT. To test this, we generated stable RPE1 cell lines in wild type and RanGAP1^ΔNES backgrounds that express fluorescent

**Fig. 6 | The N-terminal helix of RanGAP1 encodes a nuclear export signal.**
**A** Schematics for the N-terminal fragments of RanGAP1 fused to monomeric EGFP (wild type left panel; mutant with four of five NES residues mutated to alanine right panel) shown above confocal microscopy images of stable cell lines as representative z-stack average projections to highlight cytoplasmic localization for wild type and nuclear localization for the mutant. Cells stained with DAPI; GFP visualized via native GFP fluorescence using 488 nm excitation. Brightness/contrast adjusted for individual channels. **B** Schematic of CRISPR-Cas9 strategy to generate an in-frame deletion of RanGAP1 residues 5–17 to remove the RanGAP1 NES. **C** Representative Western blot (n ≥ 3 biological replicates) of wild type and clones (1, 2, and 3) expressing RanGAP1$^{\Delta NES}$ to highlight expression levels and amounts of unmodified and SUMO1-modified RanGAP1. 20 μg of lysate was loaded in each lane and probed with a rabbit polyclonal α-RanGAP1 antibody (Sigma, HPA062034) at a 1:4000 dilution (top blot) and β-actin as a loading control (bottom blot) probed with a mouse monoclonal α-β-actin antibody (Santa Cruz, SC81178) at a 1:500 dilution. **D** Representative Western blot (n = 3 biological replicates) of wild type RanGAP1 or RanGAP1$^{\Delta NES}$ lysates to highlight similar levels of phosphorylation after nocodazole treatment. 7.5 μg of lysate was loaded in each lane and blots were probed with rabbit polyclonal α-RanGAP1 antibody (Sigma, HPA062034) at 1:2000 dilution. **E** Confocal microscopy images for cells expressing wild type RanGAP1 or RanGAP1$^{\Delta NES}$ to highlight RanGAP1 localization to the nuclear periphery in both cell lines independent of the RanGAP1 NES. White arrowheads point to prototypical nuclear ring staining for RanGAP1 in the nuclear focal plane (top panels, z-stack slice shown) and RanGAP1 puncti on the periphery of the nuclear envelope after translocating 1 μm in the z-plane (bottom panels, z-stack slice shown). RanGAP1 detected with rabbit polyclonal α-RanGAP1 primary antibody (Bethyl, A302-027A) and α-rabbit IgG-Alexa647 secondary VHH nanobody (Thermo, SA5-10327), both at 1:1000 dilution. Brightness/contrast adjusted for individual channels. **F** Confocal microscopy images for cells expressing wild type RanGAP1 or RanGAP1$^{\Delta NES}$ (clones 1 and 2) in the absence or presence of leptomycin B (LMB) to highlight nuclear

accumulation of RanGAP1$^{\Delta NES}$ relative to wild type RanGAP1. Cells were treated with methanol control (left panels, -) or 50 nM leptomycin B (LMB, right panels, +) for 20–21 h, fixed, and stained for RanGAP1 using the same antibodies and dilutions as in (**E**). Representative average 2D projections of confocal z-stacks shown with brightness/contrast adjusted separately for each channel. Right graph depicts the ratio of nuclear to cytoplasmic RanGAP1 calculated using CellProfiler[83]. Individual values and bars representing means +/- SD are shown. Kruskal-Wallis with Dunn's multiple comparison tests were used to calculate statistical significance using Prism. Number of cells analyzed across at least three biological replicates was n = 110 (WT, no LMB), n = 73 (Δ1, no LMB), n = 95 (Δ2, no LMB), n = 66 (WT + LMB), n = 87 (Δ1 + LMB), and n = 72 (Δ2 + LMB). * P < 0.05, ** P < 0.01; *** P < 0.001; **** P < 0.0001; ns – not significant. See Source Data for exact p values. **G** Representative Western blot analysis of Ran (n ≥ 3 biological replicates) in cells expressing wild type RanGAP1 or RanGAP1$^{\Delta NES}$ to highlight similar levels of expression. Ran (top blot) detected with a mouse monoclonal α-Ran antibody (BD, 610340) at 1:8000 dilution and β-actin (bottom blot) as loading control (bottom blot) detected with a mouse monoclonal α-β-actin antibody (Novus, NBP1-47423) at 1:300,000 dilution. 20 μg of lysate was loaded in each lane. **H** Confocal microscopy images with an α-Ran antibody in cells expressing wild type RanGAP1 or RanGAP1$^{\Delta NES}$ (clones 1 and 2) in the absence or presence of leptomycin B (LMB) to highlight cytoplasmic mislocalization of Ran in cells expressing RanGAP1$^{\Delta NES}$ with or without LMB or in wild type cells after LMB treatment. Ran detected with mouse α-Ran monoclonal primary antibody (BD, 610340) and α-mouse IgG-Alexa647 secondary antibody (H + L Thermo A32787) at 1:250 and 1:1000 dilutions, respectively. LMB treatment, imaging, and analysis as in (**F**). At least 3 biological replicates analyzed with statistical analysis performed and indicated as in (**F**). No significant differences between the medians of Δ1 (-LMB), Δ2 (-LMB), WT ( + LMB), Δ1 ( + LMB), Δ2 ( + LMB). See Source Data for exact p-values. Number of cells analyzed was n = 112 (WT, no LMB), n = 119 (Δ1, no LMB), n = 84 (Δ2, no LMB), n = 80 (WT + LMB), n = 78 (Δ1 + LMB), and n = 73 (Δ2 + LMB).

reporters tagged with constitutive or optogenetically-inducible NES and NLS signals to quantify steady-state NCT as well as transport kinetics, respectively (Supplementary Fig. 6)[61–63]. Confocal microscopy with live cells showed no transport defects either at steady-state (Supplementary Fig. 6A–E) or with respect to the kinetics of export (Supplementary Fig. 6F–I) or import (Supplementary Fig. 6J–M) using these particular NES and NLS engineered reporter systems.

## Cells expressing RanGAP1$^{\Delta NES}$ do not exhibit mitotic defects
The RanBP2/SUMO1-RanGAP1/Ubc9 complex remains intact during mitosis after NPC disassembly and is recruited to kinetochores and mitotic spindles in a Crm1-dependent manner where the complex is thought to mediate mitotic functions involving stabilization of microtubule-kinetochore interactions as well as SUMO conjugation of topoisomerase IIα, Aurora B kinase, and other substrates[41–43,46,47,60,64–66]. Further, addition of LMB to asynchronous cells inhibits mitotic recruitment of RanBP2/SUMO1-RanGAP1/Ubc9 suggesting involvement of the Crm1 NES binding site in this process[47]. Given that RanGAP1 contains an NES, we speculated that interactions between the RanGAP1 NES and Crm1 may contribute to RanBP2/SUMO1-RanGAP1/Ubc9 recruitment by Crm1. Cells expressing RanGAP1$^{\Delta NES}$ were analyzed for differences in co-localization or defects in mitosis. However, RanGAP1$^{\Delta NES}$ co-localized with Crm1 in metaphase and anaphase in a manner similar to wild-type RanGAP1 (Supplementary Fig. 7A). Differences in mitotic localization of Ran and RanBP2 were also not observed in cells expressing RanGAP1$^{\Delta NES}$ compared to WT RanGAP1 (Supplementary Fig. 7B, C). Finally, cell cycle distributions of asynchronous cells expressing wild type or RanGAP1$^{\Delta NES}$ cells did not reveal significant differences in any cell cycle phase (Supplementary Fig. 7D, E). These results suggest that interactions between the RanGAP1 NES and Crm1 are dispensable for growth under the conditions tested and that the RanGAP1 NES is not required for Crm1-dependent recruitment of the RanBP2/SUMO1-RanGAP1/Ubc9 complex in mitosis. It is also noteworthy that SUMO1 conjugation to RanGAP1 is not essential for viability or mitotic progression in several human cell lines[67], suggesting

that SUMO1 modification of RanGAP1 and the NES might act redundantly to promote interactions between RanBP2, RanGAP1 and Crm1. Finally, we note that the RanGAP1 NES is not required for the formation of the RanBP2$^{RBD4}$/SUMO1-RanGAP1/Ubc9/Crm1/Ran$^{Q69L}$(GTP) complex in vitro (Supplementary Fig. 1D–F), consistent with previous in vitro results showing that Crm1 binding to the RanBP2$^{RBD3-4}$/SUMO1-RanGAP1/Ubc9/Ran complex does not depend on the presence of the RanGAP1 catalytic GAP domain[17].

## E3 ligase activity depends on IR1 and is modulated by Ran, Crm1 and the RanGAP1 NES
RanBP2 IR1-M-IR2 and isolated IR1 and IR2 motifs bind SUMO and Ubc9 to catalyze E3 SUMO ligase activity in vitro, with IR1 exhibiting greater activity[27,30,40,43]. While IR1 and IR2 can each bind SUMO1-RanGAP1/Ubc9, IR1 was proposed as the preferred binding site due to its higher affinity for SUMO and Ubc9[27,40,43]. This creates an apparent paradox: are these motifs receptors for SUMO1-RanGAP1/Ubc9, E3 ligases for activated SUMO-Ubc9 thioesters, or can these motifs accommodate both? One model for how the two IR modules cooperate in the context of larger RanBP2 fragments suggests that IR1 is the principal E3 ligase in the absence of SUMO1-RanGAP1/Ubc9 and that IR2 catalyzes E3 ligase activity when SUMO1-RanGAP1/Ubc9 binds and occludes IR1[43]. Our structure reveals that SUMO1-RanGAP1/Ubc9 associates with the RanBP2 IR1 motif in the context of a larger RanBP2 fragment (Fig. 7A). Further, EM densities are not apparent for IR2, suggesting high mobility, an observation consistent with the model that it binds charged SUMO - E2 for conjugation to substrates.

To test if IR1 and IR2 are required for E3 ligase activity in intact RanBP2/SUMO1-RanGAP1/Ubc9/Crm1/Ran complexes we performed SUMO2 conjugation assays with wild type RanBP2$^{RBD3-4}$ or variants containing deletions of motifs I and II in IR1 or IR2, which are required for E3 ligase activity[27]. Borealin is a component of the chromosome passenger complex (CPC) and an endogenous substrate of RanBP2 that is preferentially modified by SUMO2/3 during mitosis[41]. We reconstituted a subcomplex of the CPC with Borealin, Survivin and a

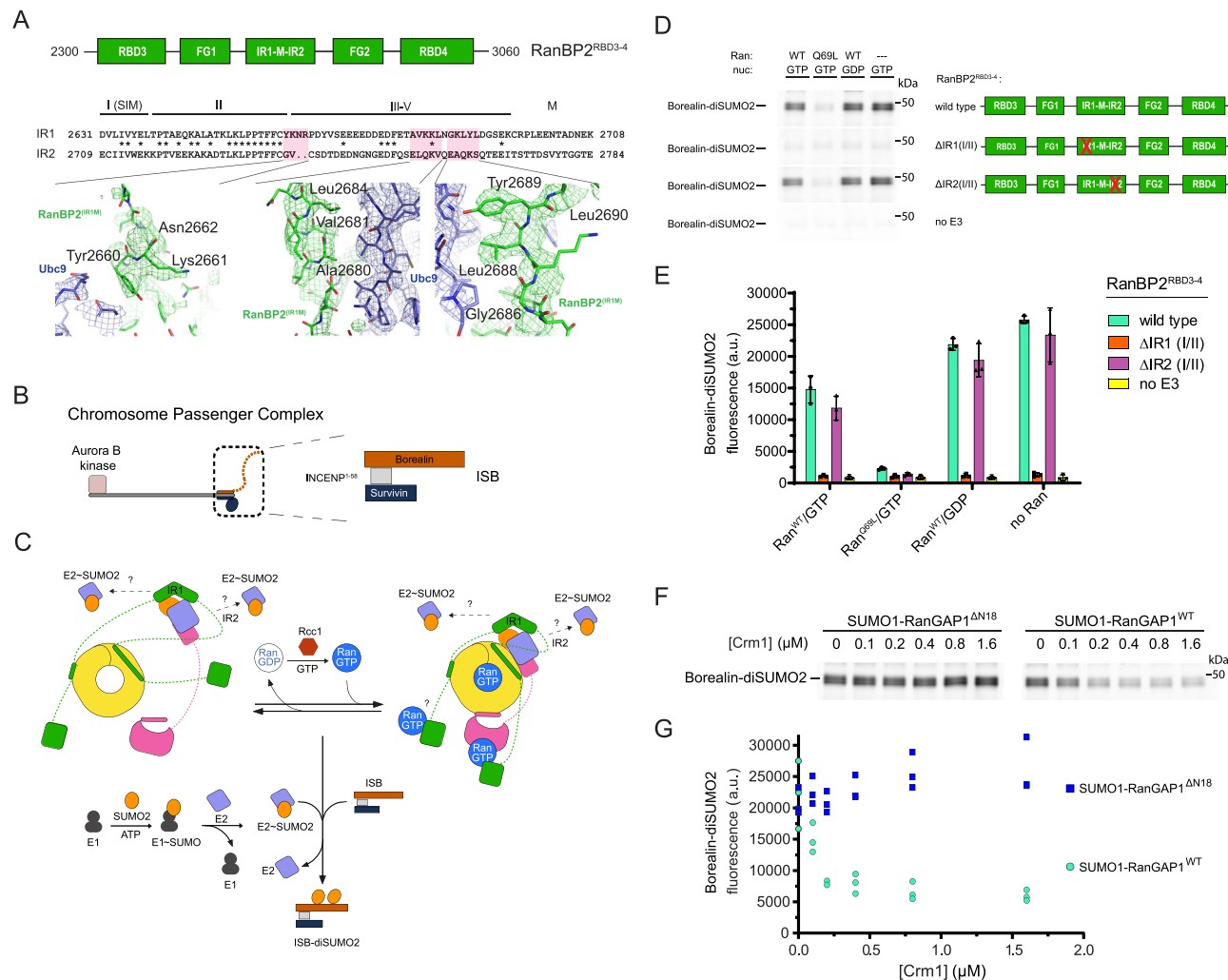

**Fig. 7 | RanBP2 E3 ligase activity depends on IR1 and cycling Ran GTPase and is inhibited by RanGAP1 NES-Crm1 interactions. A** Schematic (top) of RanBP2[RBD3-4] above the sequence alignment of IR1-M to IR2 (middle) and magnified views of highlighted regions to illustrate electron density for key residues that are present in IR1 but absent in IR2 (bottom, coloring as in Fig. 2), confirming that IR1 is the module seen in the structure. Motifs I(SIM), II–V and M[27] labeled as shown. **B** Schematic of the Chromosome Passenger Complex (CPC) indicating the full length complex containing Aurora B kinase (left) and the truncated complex used in SUMO modification assays of Borealin that lacks the Aurora B kinase but retains interactions between Incenp[1–58], Survivin, and Borealin (right)[43,85]. **C** Schematic illustrating the components and reaction intermediates in our reconstituted system for SUMO2 modification of Borealin (**D–G**) to highlight RanGAP1[NES]-free (left) and RanGAP1[NES]-bound Crm1 (right) in concert with interactions with Ran, and to highlight IR1 and IR2 as potential E3 ligases with binding of charged E2 - SUMO2 as indicated by question marks. The SUMO conjugation cascade and SUMO modification of Borealin at the bottom. RCC1 and GTP indicated and used to regenerate Ran(GTP) for Ran GTP-GDP cycling. **D** SDS-page analysis of Borealin-diSUMO2 products with variations of reaction components labeled at the top and various RanBP2 variants used indicated in schematics (right). Gels for Borealin-diSUMO2 products for wild type, ΔIR1(I/II), and ΔIR2(I/II) were cropped from a single representative gel (see Source Data); the "no E3" control was cropped from a separate gel run at the same time. Gels visualized via Alexa488 fluorescence (see Methods for details). **E** Graph showing quantification of intensities for Borealin-diSUMO2 bands from (**D**) via densitometry. Individual data points for three independent experiments shown. Bars represent means +/- SD. **F** SDS-page analysis of Borealin-diSUMO2 product formation with SUMO1-RanGAP1[ΔN18] or wild type SUMO1-RanGAP1 in the presence of wild type RanBP2[RBD3-4], Ran[Q69L](GTP), Ubc9 and increasing concentrations of Crm1 (See Methods for details). Borealin-diSUMO2 products visualized as in (**D**). **G** Graph showing quantification of intensities for Borealin-diSUMO2 bands from (**F**) via densitometry. Individual data points shown for three independent experiments.

fragment of Incenp[1–58] for use in SUMO conjugation assays (Fig. 7B)[43]. To determine if the Ran GTPase cycle modulates RanBP2 E3 ligase activity, SUMO conjugation assays were performed under multiple turnover conditions in the presence of RCC1 and either Ran(GTP), Ran[Q69L](GTP) in which GTP cannot be hydrolyzed, Ran(GDP), or with no Ran present (Fig. 7C–E, Supplementary Fig. 8). Complexes containing wild type RanBP2[RBD3-4] catalyzed SUMO2 conjugation to Borealin whereas deletion of IR1 motifs I and II resulted in a loss of Borealin modification by SUMO2 (Fig. 7D, E). By comparison, deletion of IR2 motifs I and II resulted in conjugation activity comparable to wild-type RanBP2[RBD3-4]. These results suggest that IR1 functions as the dominant

SUMO E3 ligase under these conditions. Addition of Ran(GDP) had little impact on SUMO2 modification of Borealin, but conjugation was impaired when Ran(GTP) was added under conditions where RCC1 promoted GDP/GTP exchange (Fig. 7D, E, Supplementary Fig. 8B). To further determine if Ran(GTP) cycling affects SUMO conjugation activity, Ran[Q69L](GTP) was added resulting in a substantial loss of conjugation activity suggesting that locking Ran in the GTP-bound conformation severely impairs the E3 ligase activity of the complex (Fig. 7D, E).

Ran[Q69L](GTP) inhibited conjugation in these assays, so we next explored if this was dependent on the RanGAP1 NES as this interaction

is Crm1 and Ran$^{Q69L}$(GTP)-dependent. Crm1 was titrated into reactions in the presence of excess Ran$^{Q69L}$(GTP) and RanBP2/SUMO1-RanGAP1/Ubc9 complexes containing either wild type SUMO1-RanGAP1 or SUMO1-RanGAP1$^{\Delta N18}$. Increasing Crm1 concentrations diminished SUMO conjugation activity in the presence of wild type SUMO1-RanGAP1 but no effect was observed for complexes containing SUMO1-RanGAP1$^{\Delta N18}$ that lack the NES (Fig. 7F, G). Interestingly, in the presence of leptomycin B, which is expected to block NES-Crm1 interactions, no inhibitory effect was observed for complexes with wild type SUMO1-RanGAP1 (Supplementary Fig. 8C), thus mimicking the results obtained with SUMO1-RanGAP1$^{\Delta N18}$ in the absence of LMB.

The inhibitory effect of adding Crm1 in the presence of Ran$^{Q69L}$(GTP) was not evident when using Ran(GTP) in the presence of GAP and GEF effectors (Supplementary Fig. 8A), although addition of Ran(GTP) resulted in slight inhibition of ligase activity when Crm1 was present (Fig. 7D, E and Supplementary Fig. 8B). Crm1-dependent inhibition of ligase activity in the presence of Ran$^{Q69L}$(GTP) suggests that our structure of RanBP2/SUMO1-RanGAP1/Ubc9/Crm1/ Ran$^{Q69L}$(GTP) represents an inhibited conformation for the RanBP2 E3 ligase, perhaps because IR1 is less able to dissociate from SUMO1-RanGAP1/Ubc9 to accommodate charged SUMO - E2 when the NES of RanGAP1 is stably bound to Crm1/Ran$^{Q69L}$(GTP).

## Discussion

The RanBP2/SUMO1-RanGAP1/Ubc9 complex is found at the cytoplasmic face of the NPC in interphase cells where it plays critical roles in nucleocytoplasmic trafficking[10,12,16,17]. In mitotic cells, RanBP2/ SUMO1-RanGAP1/Ubc9 remains intact and is recruited to kinetochores and spindles in a Crm1-dependent manner where it plays roles in the fidelity and progression of mitosis[42,46,47,64-66]. Importantly, RanBP2/ SUMO1-RanGAP1/Ubc9 functions as a SUMO E3 ligase targeting several substrates including a number of mitotic and interphase targets[41,42,65,68-70]. Here, we capture a structure of a RanBP2 fragment that includes its SUMO E3 ligase motifs, two FG-binding regions, and the last Ran-binding domain in complex with SUMO1-RanGAP1/Ubc9, Crm1, and Ran$^{Q69L}$(GTP). Reconstructions and models for this complex reveal interactions that were not anticipated, including identification of a RanGAP1 NES and interactions between SUMO1-RanGAP1/Ubc9 and Crm1. Our structures, combined with other recent human structures[11,57], enabled comparative analysis of interactions between the GAP domain of RanGAP1 and the switch II region of Ran, a region critical for Ran GTPase activity in the absence of a classical Arg finger. Finally, our description of the interactions between two RanBP2 FG regions, FG1 and FG2, and Crm1 expands our understanding of how karyopherins, such as Crm1 navigate the FG mesh of the NPC, while confirming several previous models of Crm1 FG binding[51,53,54].

Prior work in yeast and human cells has shown that RanGAP1 shuttles between the nucleus and cytoplasm and that human RanGAP1 accumulates in the nucleus upon treatment with LMB, suggesting Crm1-dependent export[14,44,45]. We identify the NES of RanGAP1 that contributes to its Crm1-dependent cellular localization. We also find that mislocalization of RanGAP1 to the nucleus is correlated with the mislocalization of Ran to the cytoplasm. Although the reasons for this correlation are not clear, it is possible that the generation of Ran(GDP) in the nucleus by mislocalized RanGAP1 results in unregulated diffusion of free Ran(GDP) out of the nucleus because Ran(GDP) does not bind export receptors compared to the regulated export of Ran(GTP) bound to export receptors under normal conditions. Alternatively, the generation of Ran(GDP) in the nucleus may result in reassociation with its dedicated transport factor NTF2, which only binds Ran(GDP), and their concomitant exit into the cytoplasm. Although we were unable to observe defects in NCT for several model proteins carrying classical NLS and NES signals, we speculate that the strong import and export signals in our overexpressed constructs mask defects in NCT under the conditions tested, although we cannot exclude the possibility of compensatory mechanisms that normalize NCT in the presence of mislocalized RanGAP1 and Ran in cell lines expressing RanGAP1$^{\Delta NES}$. Further work is required to determine if mislocalization of Ran or RanGAP1 contributes to NCT defects and/or differential gene expression by limiting or enhancing import and export of relevant factors, whether protein or RNA. Additionally, RanGAP1 itself may impact nuclear processes, such as heterochromatin assembly and gene expression, as has been shown in *S. pombe*[71], suggesting that mislocalization of RanGAP1 has the potential to perturb gene expression.

We observe that the IR1 SUMO E3 module of RanBP2 is bound to SUMO1-RanGAP1/Ubc9 whereas density for the IR2 module is not observed, a finding consistent with prior studies showing that IR1 forms a more stable complex and is the preferred binding partner for SUMO1-RanGAP1/Ubc9[27,40,43], Our SUMO conjugation assays show that IR1 is required and that IR2 is dispensable for SUMO2 modification of Borealin, at least in the context of the RanBP2/SUMO1-RanGAP1/Ubc9/ Crm1/Ran complex. It remains possible that IR2 may function as an E3 ligase under other conditions or for different SUMO substrates as previously put forward by Melchior and colleagues[43]. That Ran$^{Q69L}$(GTP) inhibits SUMO E3 ligase activity in the presence of Crm1 in a RanGAP1 NES-dependent manner leads us to speculate that interactions between Ran(GTP) and Crm1 can modulate RanBP2 E3 ligase activity. As Ran(GTP) levels are higher proximal to mitotic chromosomes[6,72], it is possible that the RanBP2/SUMO1-RanGAP1/ Ubc9/Crm1 complex found at kinetochores has lower E3 ligase activity and that most mitotic SUMOylation by RanBP2 does not occur while it is localized to kinetochores but rather in the mitotic cytosol as has been suggested for topoisomerase IIα, for example[42]. Alternatively, RanBP2 SUMO E3 ligase activity may persist at kinetochores because the GAP activity of RanGAP1 would promote Ran GTP-GDP cycling (Fig. 7D, E; Supplementary Fig. 8A, B). Further work is needed to understand the relationship between mitotic localization and SUMO E3 ligase activity of the RanBP2/SUMO1-RanGAP1/Ubc9 complex.

## Methods

### Cloning and expression vectors for recombinant overexpression in *E. coli*

Human RanBP2$^{2300-3224}$ (RBD3 to C-terminus), RanGAP1, Crm1, and Ran were amplified from Invitrogen human cDNA libraries (tissue from lung, kidney, testes, and/or brain). RanBP2$^{2300-3060}$ (RanBP2$^{RBD3-4}$), RanBP2$^{2300-2910}$ (RanBP2$^{RBD3}$), RanBP2$^{2446-3060}$ (RanBP2$^{RBD4}$), RanBP2$^{2300-3060,\Delta2631-2660}$ (RanBP2$^{RBD3-4,\Delta IR1(I/II)}$), and RanBP2$^{2300-3060,\Delta2709-2738}$ (RanBP2$^{RBD3-4,\Delta IR2(I/II)}$) fragments were generated by using RanBP2$^{2300-3224}$ as template and subcloned into pET-28b via restriction cloning or Gibson assembly. Wild-type RanGAP1 was used as a template to generate RanGAP1$^{\Delta N18}$, and both were cloned into pET-28b. Wild-type Ran was used as a template to generate Ran$^{Q69L}$, and both were cloned into pET-28b. Wild type Crm1 was cloned into pSmt3 (pET-28b based vector with N-terminal 6xHis-Smt3 tag). The gene for human RCC1 was synthesized in pUCIDT (IDT) with flanking NdeI and XhoI restriction sites, digested with NdeI and XhoI, and cloned into pET-28b. GST-Incenp$^{1-58}$/ Survivin/His$_6$-Borealin was synthesized as a tricistronic construct in pMK-RQ-Bb (Thermo) with restriction sites flanking each protein module and T7 promoter, lac operator, and ribosome binding site sequences from the pRSF-Duet1 vector (Novagen) between GST-Incenp and Survivin and between Survivin and His$_6$-Borealin. The synthetic construct was cut out of pMK-RQ-Bb and cloned into pRSF-Duet1 between NcoI and XhoI sites. GST was then replaced by His$_6$-Trx (thioredoxin) by restriction cloning and the His$_6$ tag on Borealin was deleted via standard PCR protocols to generate the final His$_6$-Trx-Incenp$^{1-58}$/Survivin/Borealin construct in pRSF-Duet1. Full-length SUMO2 harboring the A2C mutation was generated by PCR mutagenesis and cloned into pet21a. SUMO1$^{1-97}$ harboring the Q94P mutation was generated by PCR mutagenesis and cloned into pET-28b.

Wild-type human Ubc9 (SUMO E2) was cloned into pET-28b[73], SAE1 and SAE2[1-549] (human SUMO E1[ΔC]) were cloned into pET-11c and pET-28b[73].

## Expression and purification of RanBP2

RanBP2 variants used for cryo-EM reconstitutions and analytical gel filtration runs (see Fig. 1 and Supplementary Fig. 1) were expressed in BL21(DE3) Codon Plus (RIL) (Stratagene) cells using pET-28b vectors encoding a thrombin-cleavable His$_6$-tag. For expression, cells were grown in Superbroth medium at 37 °C in shaker flasks to OD$_{600}$ of ~4–5, transferred to an ice bath for 30 min, and induced with 0.5 mM isopropyl-beta-D-thiogalactopyranoside (IPTG) in the presence of 2% ethanol. Cultures were grown overnight at 18 °C and harvested after 16 h, pellets resuspended in 50 mM Tris pH 8.0, 20% sucrose, and flash-frozen in liquid nitrogen. RanBP2 was purified by thawing bacterial cells and lysing them in 50 mM Tris pH 8.0, 20% (w/v) sucrose, 500 mM NaCl, 20 mM imidazole, 0.1% IGEPAL, 1 mM phenylmethylsulfonyl fluoride (PMSF), 2 mM β-mercaptoethanol (BME), and 1 μL benzonase (250 U, Sigma), 5 mM MgCl$_2$, and 0.5 mg/mL lysozyme. Sonication was then performed to complete lysis using a Branson sonifier (SFX550) and lysate was then centrifuged using a Beckman JA-20 rotor at 43667 × g. Supernatant was then incubated with Ni-NTA resin (QIAGEN) equilibrated in His-wash buffer (HWB) (HWB; 20 mM Tris-HCl pH 8.0, 500 mM NaCl, 10 mM imidazole, and 1 mM tris-2-carboxyethylphosphine (TCEP)), a wash was done with HWB supplemented with 50 mM KCl and 10 mM MgSO$_4$, and the protein was eluted with His-elution buffer (HEB; 20 mM Tris-HCl pH 8.0, 500 mM NaCl, 250 mM imidazole, and 1 mM TCEP). Further purification of RanBP2 was performed via SEC with a Superdex 200 HiLoad 26/600 prep grade column (Cytiva) equilibrated in 50 mM Tris pH 8.0, 500 mM NaCl, and 1 mM TCEP. Peak fractions were pooled, concentrated to ~100 μM with Amicon Ultra-15 concentrators (Millipore Sigma), and flash-frozen prior to reconstitution.

For use in SUMO conjugation assays, wild type RanBP2$^{RBD3-4}$ was expressed as described above but in the absence of cold shock and ethanol. Pellets were resuspended in 50 mM Tris pH 7.5, 500 mM NaCl, 20 mM imidazole, 0.5 mM EDTA, 0.5 mM DTT, and SigmaFAST EDTA-free protease inhibitor tablets (lysis buffer). Cells were lysed and centrifuged as described above and supernatant was applied to Ni-NTA and washed with lysis buffer without SigmaFAST tablets (wash buffer). Protein was eluted with wash buffer supplemented with 250 mM imidazole, concentrated via Amicon Ultra-15 concentrators, and applied onto a Superdex 200 HiLoad 26/600 prep grade column equilibrated in 50 mM Tris pH 7.5, 500 mM NaCl, 0.5 mM EDTA, and 1 mM DTT. Peak fractions were collected, concentrated, and dialyzed overnight at 4 °C against 50 mM Tris pH 7.5, 150 mM NaCl, 0.5 mM EDTA, 0.5 mM DTT in the presence of thrombin to cleave the His$_6$ tag. Dialyzed protein was injected onto a Mono Q 5/50 GL column (Cytiva) equilibrated in the dialysis buffer but containing 1 mM DTT. Protein was eluted in a linear gradient from 150 mM NaCl to 500 mM NaCl over 25 column volumes. Peak fractions were pooled and exchanged into 50 mM Tris pH 7.5, 300 mM NaCl, 1 mM TCEP, 0.1 mM EDTA (BP2 storage buffer) via Amicon Ultra-15 concentrators, aliquoted, and flash frozen. RanBP2$^{RBD3-4,ΔIR1(I/II)}$ and RanBP2$^{RBD3-4,ΔIR2(I/II)}$ variants that were used in SUMO conjugation assays were expressed and purified as described for RanBP2 used in cryo-EM reconstitutions. However, after the Superdex 200 gel filtration step, RanBP2$^{RBD3-4,ΔIR1(I/II)}$ and RanBP2$^{RBD3-4,ΔIR2(I/II)}$ variants were further dialyzed, cleaved with thrombin, and run on a MonoQ 5/50 GL column exactly as described for the wild type RanBP2$^{RBD3-4}$ variant that was used in SUMO conjugation assays. Peak fractions were pooled, exchanged into BP2 storage buffer, aliquoted, and flash frozen.

## Expression and purification of RanGAP1

Full-length human RanGAP1 that was used for cryo-EM reconstitutions was expressed as described above for RanBP2 that was used in cryo-EM

reconstitutions, but the cell line used was BL21(DE3) SoluBL21 cells (Genlantis) transformed with Codon Plus (RIL) plasmid (Stratagene). Lysis and purification were also as described above for RanBP2, but with a lower NaCl concentration (350 mM vs 500 mM) at all relevant steps. Peak fractions off of the Superdex 200 26/600 column were pooled, concentrated to ~300 μM, and flash frozen.

Wild type and RanGAP1$^{ΔN18}$ proteins that were used in SUMO conjugation assays were expressed as described above without cold shock or ethanol addition during expression. Cells were harvested and lysed as described above except that lysis buffer consisted of 50 mM Tris pH 7.5, 500 mM NaCl, 20 mM imidazole, 0.25 mM EDTA, 1 mM dithiothreitol (DTT), 1.5 μL per liter of culture of benzonase (250 U), and SigmaFAST EDTA-free protease inhibitor tablets. Lysate was loaded onto Ni-NTA and washed extensively with lysis buffer but without benzonase and protease inhibitors (wash buffer). Protein was eluted from Ni-NTA with wash buffer supplemented with 250 mM imidazole and loaded onto a Superdex 200 26/600 column equilibrated with 50 mM Tris pH 7.5, 350 mM NaCl, 1 mM EDTA, and 1 mM DTT. Peak fractions were collected, concentrated to ~10 mL, supplemented with thrombin to cleave the His$_6$ tag, and dialyzed against 4 L of buffer consisting of 25 mM pH Tris 7.5, 150 mM NaCl, 0.5 mM DTT, and 1 mM EDTA. Dialyzed sample was loaded onto a Mono Q 10/100 GL column (Cytiva) equilibrated with the dialysis buffer and protein was eluted via a linear gradient from 150 mM to 600 mM NaCl over 13 column volumes. Peak fractions were concentrated, exchanged into 25 mM Tris pH 7.5, 250 mM NaCl, 0.1 mM EDTA, and 1 mM TCEP via Amicon-15 Ultra concentrators, and flash frozen.

## Expression and purification of Crm1

Full-length human Crm1 that was used for cryo-EM reconstitutions was expressed as described above for RanGAP1, which was used in cryo-EM reconstitutions. A pSmt3 vector, containing an N-terminal His$_6$-Smt3 tag, was used as a solubility enhancer for expression of Crm1. Lysis was carried out with 50 mM Tris pH 8.0, 20% (w/v) sucrose, 350 mM NaCl, 20 mM imidazole, 0.1% IGEPAL, 1 mM PMSF, 2 mM BME, and 1 μL benzonase (250 U), 2 mM Mg(OAc)$_2$, and 0.5 mg/mL lysozyme. Sonication was then performed to complete lysis using a Branson sonifier and the lysate was centrifuged using the JA-20 rotor at 43667 x g. Supernatant was incubated with Ni-NTA resin equilibrated against His-wash buffer (HWB; 20 mM Tris pH 8.0, 500 mM NaCl, 10 mM imidazole, and 1 mM TCEP), a wash was done with HWB supplemented to a final concentration of 50 mM KCl and 10 mM MgSO$_4$, and the protein was eluted with His-elution buffer (HEB; 20 mM Tris pH 8.0, 500 mM NaCl, 250 mM imidazole, and 1 mM TCEP). The Ni eluate was then dialyzed O/N at 4 °C (against 50 mM Tris pH 8.0, 150 mM NaCl, 2 mM Mg(OAc)$_2$, and 1 mM TCEP) in the presence of Ulp1 (to cleave Smt3 tag). The recovered dialysate was then loaded onto a Superdex 200 HiLoad 26/600 prep grade column equilibrated with 50 mM Tris pH 7.5/8.0, 50 mM NaCl, 2 mM Mg(OAc)$_2$, and 1 mM TCEP. Peak fractions were pooled and loaded onto a MonoQ 10/100 GL equilibrated in 50 mM Tris pH 8.0, 50 mM NaCl, 2 mM Mg(OAC)$_2$, 1 mM TCEP in order to further purify Crm1 and remove any trace Ulp1. Crm1 was eluted with a linear gradient of 50 mM Tris pH 8.0, 1 M NaCl, 2 mM Mg(OAc)$_2$, 1 mM TCEP. Peak fractions were pooled, concentrated to ~90 μM with Amicon Ultra-15 concentrators, and flash frozen prior to reconstitution.

Crm1 protein used for SUMO conjugation assays and in analytical gel filtration experiments (Supplementary Fig. 1) was expressed as described above except no cold shock was done and ethanol was not added during expression. Pellets were harvested and resuspended in lysis buffer consisting of 50 mM Tris pH 7.5, 500 mM NaCl, 20 mM imidazole, 0.5 mM TCEP, 2 mM Mg(OAc)$_2$, and SigmaFAST EDTA-free protease inhibitor tablets. Cells were lysed and protein was bound to Ni-NTA as described above except the wash buffer consisted of 50 mM Tris pH 7.5, 500 mM NaCl, 20 mM imidazole, 0.5 mM TCEP, and 2 mM

Mg(OAc)$_2$ without KCl or MgSO$_4$. Dialysis and Ulp1 cleavage proceeded as above except Tris pH 7.5 was used whereas the Superdex 200 HiLoad 26/600 run was done in 50 mM Tris pH 7.5, 100 mM NaCl, 2 mM Mg(OAc)$_2$, and 1 mM TCEP. Peak fractions were pooled, diluted with buffer for a final concentration of 60 mM NaCl, and loaded onto a Mono Q 10/100 GL column equilibrated in 50 mM Tris pH 7.5, 60 mM NaCl, 2 mM Mg(OAc)$_2$, and 1 mM TCEP. Crm1 was eluted in a linear gradient from 60–300 mM NaCl over 20 column volumes. Peak fractions were collected, exchanged into 50 mM Tris 7.5, 150 mM NaCl, 2 mM Mg(OAc)$_2$, and 1 mM TCEP via Amicon Ultra-15 concentrators, and flash frozen.

### Expression and purification of Ran(GTP)

Human Ran was expressed as described above for RanBP2. Lysis of bacterial cells was performed in the presence of 50 mM Tris pH 8.0, 500 mM NaCl, 15 mM imidazole, 4 mM MgCl2, 10% glycerol, 2.5 mM β-mercaptoethanol, 1 mM PMSF, 0.5 mg/mL lysozyme, 20 μM GTP, and 1 μL benzonase (250 U). Sonication was performed to complete lysis using a Branson sonifier and lysate was centrifuged using a Beckman JA-20 rotor at 43667 x g. Supernatant was then incubated with Ni-NTA resin equilibrated with 20 mM Tris-HCl pH 8.0, 500 mM NaCl, 15 mM imidazole, 4 mM MgCl$_2$, 10% glycerol, 2.5 mM BME, and 20 μM GTP, and the protein was eluted with 20 mM Tris-HCl pH 8.0, 500 mM NaCl, 500 mM imidazole, 4 mM MgCl$_2$, 10% glycerol, 2.5 mM BME, 20 μM GTP. After elution, the protein sample was treated with thrombin to cleave the His$_6$ tag and dialyzed overnight against 20 mM Tris pH 8.0, 250 mM NaCl, 4 mM MgCl$_2$, 10% glycerol, 1 mM TCEP and 10 μM GTP. Upon recovery from dialysis, Ran was loaded with GTP nucleotide by first incubating at RT for 30 min with 10x molar excess of Tris-buffered GTP pH 7.5 and 20 mM EDTA pH 8.0. Then, Tris-buffered 1 M MgCl$_2$ pH 7.5 was added to a final concentration of 40 mM MgCl$_2$, and the sample was incubated on ice for 30 min. After this, Ran was loaded onto a Superdex 75 HiLoad 26/600 (Cytiva) prep grade column for size-exclusion chromatography. Peak fractions were pooled and dialyzed a second time, against 20 mM Tris pH 8.0, 50 mM NaCl, 5 mM MgCl$_2$, 1 mM TCEP, followed by anion exchange chromatography on a MonoQ 10/100 GL column to separate Ran(GTP) from Ran(GDP). A linear ionic strength gradient was applied from 50 mM NaCl to 1 M NaCl, in the presence of 20 mM Tris pH 8.0, 5 mM MgCl$_2$, and 1 mM TCEP. Appropriate peak fractions were pooled and concentrated in 3 K MWCO Amicon Ultra-15 concentrators, flash frozen, and stored at −80 °C.

Wild type Ran protein used for SUMO conjugation assays was expressed as described above except no cold shock was done and ethanol was not added during expression. Purification was as described above except 20-50 mM Tris pH 7.5 was used as buffer, 0.5–1 mM DTT as reducing agent, SigmaFAST EDTA-free tablets as protease inhibitor, and no lysozyme was used during lysis.

### Expression and purification of human RCC1

BL21(DE3) Codon Plus (RIL) cells harboring pET-28b-hRCC1 (containing a thrombin-cleavable N-terminal His$_6$ tag) were grown at 37 °C in LB to OD 0.6–0.8, shifted to 18 °C, and induced with 0.5 mM IPTG. After 16-20 h of expression cells were pelleted and resuspended in a lysis buffer consisting of 50 mM Tris pH 7.5, 500 mM NaCl, 20 mM imidazole, 0.25 mM EDTA, 0.5 mM DTT, 1.5 μL of Ultrapure Benzonase (Sigma) per 1 L of culture, and *Complete* EDTA-free protease inhibitor cocktail tablets (Roche). After sonication, lysate was centrifuged using a Beckman JA-20 rotor at 39,191 × g and supernatant was applied onto Ni-NTA beads. After washing the Ni-NTA beads with at least 10 CV of wash buffer (lysis buffer without protease inhibitors or Benzonase), protein was eluted with a buffer containing 50 mM Tris pH 7.5, 300 mM NaCl, 250 mM imidazole, 0.25 mM EDTA, 0.5 mM DTT. Eluted protein was concentrated via Amicon Ultra-30 concentrators and diluted 10-fold with a buffer consisting of 50 mM Tris pH 7.5, 150 mM

NaCl, 0.5 mM DTT, and 1 mM EDTA. The His$_6$ tag was cleaved by adding 50 units of thrombin (Sigma, T7513) and incubating overnight at 4 °C. After cleavage by thrombin, protein was injected onto the Superdex HiLoad S200 26/600 column equilibrated in 50 mM Tris pH 7.5, 150 mM NaCl, 1 mM DTT, and 1 mM EDTA. Peak fractions were pooled, concentrated, and injected onto a Mono S 10/100 GL column equilibrated in the same buffer as the S200 26/600 column. Protein was eluted from 150 mM NaCl – 600 mM NaCl over ten column volumes. Peak fractions were pooled, concentrated, and exchanged into 50 mM Tris pH 7.5, 200 mM NaCl, 1 mM DTT, 1 mM EDTA via an Amicon concentrator. Protein concentration was measured via absorbance at 280 nm and protein was flash frozen and stored at −80 °C.

### Expression and purification of the Incenp$^{1–58}$-Survivin-Borealin complex

BL21 DE3 Codon Plus (RIL) cells harboring pRSFDuet His$_6$-Trx-Incenp$^{1–58}$-Survivin-Borealin (His$_6$-Trx-ISB) were grown in LB at 37 °C to OD - 0.7, protein expression was induced by the addition of 0.5 mM IPTG, and cells were moved to 18 °C for overnight expression. After 16–20 h, cells were pelleted and resuspended in lysis buffer consisting of 50 mM Tris pH 7.5, 500 mM NaCl, 20 mM imidazole, 0.5 mM DTT, 0.5 mM EDTA, 1.5 μL of Ultrapure Benzonase per 1 L of culture, and 1 tablet of SigmaFAST protease inhibitor per 60 mL of buffer. Cells were lysed, centrifuged, and applied to Ni-NTA as described for RCC1 above. Ni-NTA beads were washed with at least 10 CV of wash buffer (same as for RCC1) and eluted in 50 mM Tris pH 7.5, 500 mM NaCl, 250 mM imidazole, 0.25 mM EDTA, 0.5 mM DTT. The Ni-NTA eluate was concentrated via Amicon Ultra-15 concentrator and loaded onto the Superdex HiLoad S200 26/600 column pre-equilibrated in 50 mM Tris pH 7.5, 350 mM NaCl, 1 mM DTT, and 1 mM EDTA. Appropriate fractions were pooled, concentrated to -10 mL, and TEV protease (in-house) was added to a final concentration of 0.5 μM to cleave the His$_6$-Trx tag. Sample was dialyzed overnight at 4 °C into 25 mM HEPES pH 7.5, 150 mM NaCl, 2 mM DTT, 1 mM EDTA using a 12 mL 7 K MWCO dialysis cassette (Thermo). Dialyzed protein was loaded onto the Mono S 10/100 GL equilibrated in 25 mM HEPES pH 7.5, 150 mM NaCl, 1 mM DTT, 1 mM EDTA and eluted via gradient from 150 mM NaCl – 750 mM NaCl over 15 column volumes. Appropriate fractions were concentrated and exchanged into 50 mM Tris pH 7.5, 250 mM DTT, 0.25 mM EDTA via the Amicon Ultra-15 concentrator, aliquoted, and frozen at −80 °C.

### Expression and purification of SUMO2$^{A2C}$, SUMO1$^{Q94P}$, Ubc9, and SUMO E1$^{ΔC}$

Full length C-terminally His$_6$-tagged SUMO2$^{A2C}$ and N-terminally His$_6$-tagged mature SUMO1$^{Q94P}$ were expressed in BL21 DE3 Codon Plus (RIL) cells in LB and purified as described for other His$_6$ tagged SUMO variants[33,73]. BL21 DE3 Codon Plus (RIL) cells expressing His$_6$-tagged Ubc9 were grown in LB at 37 °C until OD - 0.6-0.8, cold-shocked on ice, and shifted to 18 °C prior to adding 2% ethanol and 0.5 mM IPTG. Ubc9 was lysed and purified in a manner similar to the RanBP2 that was used for cryo-EM reconstitutions except the Ni-NTA buffer contained 350 mM NaCl and the Ni-NTA eluate was concentrated and injected onto a Superdex 75 HiLoad 26/600 column equilibrated in 20 mM Tris pH 8.0, 150 mM NaCl, and 1 mM TCEP. Appropriate gel filtration fractions were pooled, concentrated, and flash frozen. SUMO E1$^{ΔC}$ was expressed and purified as described above for Ubc9 except the Ni-NTA eluate was injected onto a Superdex 200 HiLoad 26/600 column, peak fractions were pooled, concentrated, and flash frozen in liquid nitrogen.

### Reconstitution of RanBP2 Complexes

Purified RanGAP1 was first conjugated to SUMO1$^{Q94P}$ in vitro. Briefly, RanGAP1 and SUMO1 were incubated in a 1:1.5 molar ratio for 7 h at 37 °C in the presence of human SUMO E1 and E2 enzymes in 20 mM

HEPES pH 7.4, 50 mM NaCl, 5 mM MgCl$_2$, 1 mM TCEP, and 5 mM ATP. The reaction was tracked by SDS-PAGE with hourly time points to determine when ~95% of RanGAP1 was modified by a single SUMO1 moiety. Purification of SUMO1$^{Q94P}$-RanGAP1 from leftover unreacted species was performed via anion exchange chromatography by passing over a MonoQ 10/100 GL column equilibrated in 20 mM Tris pH 8.0, 100 mM NaCl, 1 mM TCEP and eluted via a linear gradient of Buffer B, 20 mM Tris pH 8.0, 1000 mM NaCl, 1 mM TCEP. RanBP2$^{RBD4}$/RanGAP1-SUMO1$^{Q94P}$/Ubc9 was then reconstituted by incubating purified RanGAP1-SUMO1$^{Q94P}$ with excess RanBP2$^{RBD4}$ and Ubc9 (1.0: 2.5: 2.5 molar ratio), dialyzed against 20 mM Tris pH 8.0, 100 mM NaCl, 1 mM TCEP, and purified on a MonoQ column (Buffer A: 20 mM Tris pH 8.0, 50 mM NaCl, 1 mM TCEP; Buffer B: 20 mM Tris pH 8.0, 1000 mM NaCl, 1 mM TCEP) and eluted via a two-step gradient where the first was 0-30% B over 2 column volumes and the second was 30-55% B over 14 column volumes. Finally, purified RanBP2$^{RBD4}$/SUMO1-RanGAP1$^{Q94P}$/Ubc9 was incubated with excess Crm1 and Ran$^{Q69L}$(GTP) and purified by gel filtration on an S200 Increase 10/100 GL (Cytiva) column equilibrated with 20 mM HEPES pH 7.4, 110 mM potassium acetate, 2 mM magnesium acetate, and 1 mM TCEP. The appropriate fractions were pooled and the complex was concentrated to ~ 9 mg/mL using Amicon UltraCel 10 K MWCO centrifugal filters (Merck Millipore).

## Crosslinking for cryo-electron microscopy
Reconstituted and purified RanBP2$^{RBD4}$/SUMO1$^{Q94P}$-RanGAP1/Ubc9/Crm1/Ran$^{Q69L}$(GTP) was lightly crosslinked with 250 µM bis(sulfosuccinimidyl)suberate (BS$^3$), an amine-to-amine crosslinker that is water-soluble and non-cleavable. After crosslinking for 30 min, the reaction was quenched with 50 mM Tris pH 7.5. The sample was then supplemented with 0.1% CHAPSO and immediately used for grid preparation. Aliquots for analysis of protein were added to 4X LDS sample buffer supplemented with 10% v/v BME and separated by SDS-PAGE followed by Coomassie or SYPRO Ruby staining.

## Cryo-electron microscopy data collection
UltrAuFoil R 1.2/1.3 holey grids were glow-discharged for 60 s at 0.37 mBar, 15 mA in a Pelco easiGlow air system (TED PELLA). Three µL of a suspension of lightly cross-linked complexes of hRanBP2$^{RBD4}$/SUMO1$^{Q94P}$-RanGAP1/Ubc9/Crm1/Ran$^{Q69L}$(GTP) at a concentration of 9.7 mg/mL were applied per grid and plunge-frozen in liquid ethane using a Vitrobot Mark IV (FEI) at 22 °C, 100% humidity (20 s wait time, 3.0 s blot time). Grids were screened for suitable particle density using an MSKCC in-house Titan Krios 300 kV (FEI) microscope and Serial EM data collection software[74]. Data collection was performed on a Titan Krios 300 kV (FEI) instrument using a K2 Summit direct detector (Gatan). A total of 6882 movies (50 frames/movie, 10 s exposure time) were collected from 4 grids in super-resolution mode with a defocus range from −1.0 to −3.0 µm at a dose rate of 10.0 e-/px/sec and a total dose of 85.2 e-/Å$^2$/movie. The exposure of the movies was filtered in a dose-dependent manner to weigh the amplitudes of each movie frame, preserving low-resolution contrast across the exposure while favoring high-resolution information from early frames that sustained less radiation damage.

## Image processing and map reconstruction
Movies were processed with MotionCor2 to generate dose-weighted aligned and averaged micrographs alongside aligned movies with a pixel size of 1.09 Å/pixel[75]. 5964 micrographs were selected for further processing based on their appearance, CTF estimation and number of particles. In RELION 3.0[76] ~ 1000 particles were selected to generate a small set of 2D class averages that were then used as templates for auto-picking the remaining particles. During initial auto-picking in RELION 3.0, 1294521 particles were extracted with a box size of 300 × 300 pixels. This particle stack was input to cryoSPARC2 (Structura Biotechnology Inc) to obtain an initial set of 2D class averages and

to remove junk particles. After 2D classification in cryoSPARC2, 895509 particles remained. These ~895 k particles were used for generation of ab initio reconstructions in cryoSPARC2, asking in parallel for 1, 2, 3, and 4 distinct reconstructions. Initial 3D classification in cryoSPARC2 (employing heterogeneous refinements) resulted in isolation of a subset of particles (count: 540834) containing Crm1 bound with Ran$^{Q69L}$(GTP) and an amorphous protrusion of electron density off the side of Crm1 that was preliminarily thought to be the GAP domain of RanGAP1. Left behind from this classification were ~322 k particles that upon reconstruction gave rise to free Crm1. The ~540 k subset of particles clearly containing a central Crm1/Ran$^{Q69L}$(GTP) were ported back into RELION 3.0 for further analysis. After initial rounds of particle orientation assignment and image reconstruction, focused refinement of the central Crm1/Ran$^{Q69L}$(GTP) was performed, as this was observed to be the part of the structure that exhibited the greatest degree of structural rigidity. Using these orientations as a starting point, four rounds of the following four steps were carried out to improve the achievable resolution of the particle data: (1) Bayesian particle polishing, (2) 3D refinement followed by focused refinement on Crm1/Ran$^{Q69L}$(GTP), (3) defocus (CTF) refinement, and finally (4) another 3D refinement step.

Subsequently, another round of 3D refinement followed by focused refinement on Crm1/Ran$^{Q69L}$(GTP) resulted in a sharpened map resolution of 2.89 Å for the central Crm1/Ran$^{Q69L}$(GTP). Additional densities were observed adjacent to the central Crm1/Ran$^{Q69L}$(GTP). To resolve these densities, a 3D mask was generated around Crm1/Ran$^{Q69L}$(GTP) to subtract that region within each particle. These subtracted particles were 3D classified without image alignment, using a reference mask that was large enough to accommodate the volume of all possible orientations. Seven of eight classes exhibited densities, which were resolved as the GAP domain of RanGAP1 bound to Ran$^{Q69L}$(GTP), which was engaging the RBD4 of RanBP2. Each of these were masked to subtract densities outside the mask followed by recentering as implemented in RELION 3.1.2. The resulting GAP/Ran$^{Q69L}$(GTP)/RBD4 particles were aligned to a 3D reference and following several rounds of focused 3D classification without image alignment, the final, postprocessed reconstruction of the RanGAP1$^{GAP}$/Ran$^{Q69L}$(GTP)/RanBP2$^{RBD4}$ subcomplex was resolved to a resolution of 3.4 Å followed by focused refinements.

The third area of the structure encompassing the SUMO1-RanGAP1$^{CTD}$/Ubc9/RanBP2$^{IR1}$ subcomplex was determined using a similar strategy whereby regions of the map not encompassing this subcomplex were subtracted from all particles, followed by a 3D focused classification with no image alignment, to isolate all particles that included the SUMO1-RanGAP1$^{CTD}$/Ubc9/RanBP2$^{IR1}$ subcomplex. The final postprocessed maps for the SUMO1-RanGAP1$^{CTD}$/Ubc9/RanBP2$^{IR1}$ subcomplex were resolved to 3.29 Å and 3.1 Å resolution after focused refinements.

## Model building and refinement
Components of atomic models for the RanBP2$^{RBD4}$/SUMO1-RanGAP1/Ubc9/Crm1/Ran$^{Q69L}$(GTP) complex were docked into individual maps and manually rebuilt with Coot[77] using a prior crystal structure of human SUMO2-RanGAP1$^{CTD}$/Ubc9/RanBP2$^{IR1}$ (PDB 3UIN), human Snurportin1 and Ran(GTP) and mouse Crm1 (PDB 3GJX), human RanBP2 RBD1 (PDB 4L6E), and the human Ran(GMPPNP) complex with human RanBP1 and the *Schizosaccharomyces pombe* GAP domain of RanGAP1 (PDB 1K5D). These models were used with all of the aforementioned individual reconstructions to generate a composite map using Phenix[78]. Elements of RanBP2, SUMO1 and other non-human components from docked crystal structures were then manually placed or rebuilt into the composite map using Coot and refined using Phenix. Model geometry was analyzed using Molprobity[79]. Local resolution maps were generated using Phenix. Structures and maps were rendered using ChimeraX[80] or Pymol (Schrödinger, LLC).

Sphericity for EM reconstructions calculated using 3DFSC using masks at FSC = 0.5[81].

## Analytical gel filtration of individual components and complexes

Purified individual components (His$_6$-RanBP2 variants, His$_6$-Ubc9, SUMO1-RanGAP1, Crm1, and Ran$^{(Q69L)}$(GTP)) and reconstituted RanBP2/SUMO1-RanGAP1/Ubc9 variant subcomplexes (purified via ion exchange as described for RanBP2$^{RBD4}$/RanGAP1-SUMO1$^{Q94P}$/Ubc9 above) were incubated alone or in combination for 15 min on ice at the indicated concentrations in 20 mM HEPES pH 7.4, 110 mM potassium acetate, 2 mM magnesium acetate, and 1 mM TCEP. 50 µL of sample was then injected onto a Superdex 200 Increase 3.2/300 (Cytiva) equilibrated in the same buffer and the column was run at 4 °C at 0.075 mL/min. 100 µL fractions were collected for SDS-PAGE analysis.

## SUMO-conjugation assays

Wild type RanGAP1 and RanGAP1$^{ΔN18}$ were purified as described above for SUMO conjugation assays and conjugated to SUMO1$^{Q94P}$ largely as described in the "Electron Microscopy Sample Preparation, Imaging and Analysis" section except the conjugation was allowed to proceed for 18 h at 37 °C and Mono Q 10/100 purification was done in 50 mM Tris pH 7.5, 1 mM DTT, 0.5 mM EDTA using a linear gradient of 250 mM to 600 mM NaCl over 15 column volumes to elute the protein. SUMO2$^{1-93,A2C}$ was conjugated to Alexa488-maleimide[33].

For conjugation reactions with RanBP2$^{RBD3-4}$, RanBP2$^{RBD3-4,ΔIR1(I/II)}$, and RanBP2$^{RBD3-4,ΔIR2(I/II)}$, 300 nM Ubc9 (E2), 100 nM Aos1/Uba2$^{ΔC}$ (E1), 5 µM SUMO2$^{1-93,A2C}$-Alexa488, 2 µM Incenp$^{1-58}$-Survivin-Borealin (ISB), 500 nM RCC1, 200 nM SUMO1$^{Q94P}$-RanGAP1, 750 nM Crm1, 100 nM RanBP2 variant (E3), 500 nM Ran variant (when present), and 3 mM GTP or GDP were incubated on ice in Buffer K (20 mM HEPES pH 7.5, 100 mM KOAc, 5 mM MgCl$_2$, 1 mM TCEP) and reaction was started by the addition of 1 mM ATP and transferred to a 37 °C heat block. After 50 min at 37 °C, 8 µL of sample was quenched with 4 µL of 4x NuPAGE LDS (Thermo) containing DTT. 6 µL of quenched reaction was run on 4–12% NuPAGE Bis-Tris gel (Thermo), and the gel was scanned with Typhoon FLA-9500 (Cytiva) at 500 V using the 473 nm laser line. Intensity of bands corresponding to SUMO2-conjugated Borealin were quantified with ImageJ and plotted using Prism (GraphPad). To test the effect of Crm1 on E3 ligase activity of RanBP2$^{RBD3-4}$ in the presence or absence of the N-terminus of RanGAP1 or of 2 µM leptomycin B, reactions were performed essentially as described above except 3.2 µM Ran$^{Q69L}$(GTP) or 3.2 µM wild type Ran was used, Crm1 was titrated starting at 1.6 µM via 2-fold serial dilutions, and reactions were run at 37 °C for 96 min for Ran$^{Q69L}$(GTP) experiments and for 50 min for wild type Ran assays. Gel shown in Supplementary Fig. 8C was scanned at 425 V on Typhoon 5 (Amersham) using 488 nm laser for excitation and the Cy2 emission filter.

## Vectors, primers, and guide RNAs used for CRISPR

pSpCas9(BB)−2A-GFP (PX458) was a gift from Feng Zhang (Addgene plasmid # 48138; http://n2t.net/addgene:48138; RRID:Addgene 48138). PX458 harboring mCherry instead of EGFP (PX458-mCherry) was constructed by digesting parental PX458 with EcoRI and ligating a T2A-mCherry gBlock (IDT) also digested with EcoRI. Guide RNAs with the sequences CAGCTTGGCAATGTCTTCCG (sgRNA236, target site ten base pairs from A of the initiator ATG codon of the RanGAP1 gene) and GACACTTGCCAAGACTCAGG (sgRNA240, target site 49 base pairs from A of the initiator ATG codon of the RanGAP1 gene) where generated with BbsI sites, annealed, and cloned into PX458 and PX458-mCherry using the BbsI restriction site following Zhang lab general cloning protocol (https://media.addgene.org/cms/filer_public/6d/d8/6dd83407-3b07-47db-8adb-4fada30bde8a/zhang-lab-general-cloning-protocol-target-sequencing_1.pdf). PCR amplification of genomic DNA for verification of CRISPR-mediated deletion

was done with primers VB243 (TACTGCAACTTTGGCCTCCT) and VB244 (GGGCAAATGATCTCTGAGCC) which bind upstream of exon 1 and between exons 1 and 2, respectively.

## Transfection with CRISPR-Cas9 plasmids and selection of single clones

Human hTERT-immortalized retinal pigment epithelial cells (hTERT-RPE1; ATCC, CRL-4000) were maintained at 37 °C and 5% CO$_2$ in regular media (DME:F12 HG containing 15 mM HEPES, 2.4 mg/mL NaHCO$_3$, 2.5 mM L-glutamine (MSK Media Preparation Core)) supplemented with 10% FBS (Sigma, F2442) and 1% penicillin/streptomycin (vol/vol; Gibco). Prior to transfection, cells were moved to antibiotic-free media. At ~70% confluency, one 10 cm plate of hTERT-RPE1 cells was co-transfected with 10 µg of PX458 harboring sgRNA236 and 10 µg of PX458-mCherry harboring sgRNA240 using 43.4 µL Lipofectamine 3000 and 40 µL P3000 reagent following the Lipofectamine 3000 protocol (Thermo). Three days post transfection cells were trypsinized, resuspended in FACS-sorting media (1x PBS with 25 mM HEPES pH 7.0, 2% FBS, 2 mM EDTA), and sorted on an Aria flow cytometer (BD, MSK Flow Cytometry Core Facility) selecting for top GFP and mCherry co-expressing cells (~5600 cells; about 12% of FACS input). Sorted cells were maintained in regular media with 1% pen/strep for 3 weeks then dilution cloned on 24-well plates at 0.5 cells/well. Single clones were expanded to 6-well plates after ~2 weeks of growth on 24-well plates and genomic DNA was extracted using Quick Extract DNA extraction solution (Lucigen). Primers flanking the expected deletion region (VB243 and VB244) were used to screen for clones with deleted RanGAP1 residues 5–17 (a ~ 40 base pair deletion compared to wild type). PCR products corresponding to the size of the expected 40 base pair deletion were gel extracted and cloned into CloneJet (Thermo) for sequencing. Western blots on whole cell lysates with several α-RanGAP1 antibodies (Sigma HPA062034, Santa Cruz sc-28322) were done on each clone to confirm the expression of truncated RanGAP1.

## Cloning of constructs for the generation of stable hTERT-RPE1 cell lines

Full length wild type RanGAP1 or full length RanGAP1 containing the I6A, L9A, L13A, T16A mutations (4 A mutant) were cloned on the N-terminus of monomeric EGFP in the mEGFP-N1 vector (mEGFP-N1 was a gift from Michael Davidson - Addgene plasmid # 54767; http://n2t.net/addgene:54767; RRID:Addgene_54767) via restriction site cloning (BglII, KpnI sites). PCR was then used to delete residues 19-587 of RanGAP1 to generate RanGAP1$^{1-18-WT}$-mEGFP and RanGAP1$^{1-18-4AMUT}$-mEGFP. To generate pEGFP-N1-2xGFP:NES-IRES-2xRFP:NLS, pLVX-EF1alpha-2xGFP:NES-IRES-2xRFP:NLS was used as template for PCR addition of NheI and XbaI sites to the 2xGFP:NES-IRES-2xRFP:NLS insert (pLVX-EF1alpha-2xGFP:NES-IRES-2xRFP:NLS was a gift from Fred Gage; Addgene plasmid # 71396; http://n2t.net/addgene:71396; RRID:Addgene_71396), and the insert was cut out and ligated into pEGFP-N1 plasmid prepped from a *dam$^-$ dcm$^-$ E.coli* strain (New England Biolabs Cat# C2925H) and digested with NheI and XbaI. NLS-mCherry-LEXY (pDN122) was a gift from Barbara Di Ventura & Roland Eils (Addgene plasmid # 72655; http://n2t.net/addgene:72655; RRID:Addgene_72655). NES-mCherry-LiNUS (pDN77) was a gift from Barbara Di Ventura & Roland Eils (Addgene plasmid # 61347; http://n2t.net/addgene:61347; RRID:Addgene_61347).

## Transfection and generation of stable cell lines

hTERT-RPE1 cells were maintained in DME:F12 HG media containing 15 mM HEPES, 2.4 mg/mL NaHCO$_3$, 2.5 mM L-glutamine supplemented with 10% FBS and 1% penicillin/streptomycin as described above. To generate stable hTERT-RPE1 cell lines, 2.5 µg of plasmid was transfected into the appropriate hTERT-RPE1 cell line at ~70–90% confluency following the Lipofectamine 3000 manufacturer's protocol.

After 5–7 days, cells were moved into the same DME:F12 media but supplemented with 300 µg/mL G418 (Sigma) instead of 1% pen/strep. After 16–17 days in selection media, aliquots of cells were frozen down in Recovery Cell Culture Medium (Gibco) and passaging was continued in regular media supplemented with 200 µg/mL G418 in the absence of pen/strep. hTERT-RPE1 cells expressing 2xGFP:NES-IRES-2xRFP:NLS were further sorted by FACS on an Aria flow cytometer (MSK Flow Cytometry Core Facility) for the top-expressing GFP+/RFP+ double positive cells using the same gating for wild type and RanGAP1$^{\Delta NES}$ expressing cell lines.

### Leptomycin B treatment and fixation for confocal microscopy for RanGAP1 and Ran N/C localization analyses

Wild type RPE1 and RanGAP1$^{\Delta NES}$ cells were plated on 4-well µ-slides (Ibidi), allowed to attach overnight, and 50 nM leptomycin B (LMB, Enzo Life Sciences) was added for 20–21 h where indicated. Cells were washed twice with 1x PBS containing 1 mM MgCl$_2$ and 1 mM CaCl$_2$ (PBS$^{++}$) and fixed in 2% paraformaldehyde (PFA, Biotium) in PBS for 20 min at room temperature. After fixation, cells were washed three times in 1x TBS containing 0.05% Tween-20 (TBST, Thermo) and then permeabilized in 100% methanol for 20 min at −20 °C. Cells were washed three times with ice cold TBST and incubated with TBST containing 2% BSA (TBSTB) for 30 min at room temperature. For RanGAP1 localization experiments, rabbit α-RanGAP1 primary antibody (Bethyl, A302-027A) was diluted 1:1000 in TBSTB. For Ran localization experiments, mouse α-Ran primary antibody, (BD, 610340) and rabbit α-RanGAP1 primary antibody (Bethyl, A302-027A) were diluted 1:250 and 1:1000, respectively, in TBSTB. The primary antibodies were added to cells and incubated overnight at 4 °C with mild agitation. Cells were washed three times with TBST for 5 min each at room temperature. For RanGAP1 localization studies, alpaca α-rabbit-IgG secondary nanobody conjugated to Alexa-647 (Thermo SA5-10327) was diluted in TBSTB at 1:1000 and incubated with cells for 30 min at room temperature. For Ran localization experiments, alpaca anti-rabbit IgG secondary nanobody conjugated to Alexa488 (VHH, Thermo SA5-10323) and donkey anti-mouse IgG secondary antibody conjugated to Alexa647 (H + L Thermo A32787) were each diluted in TBSTB at 1:1000 and added to cells for 30 min at room temperature. Cells were washed again in TBST three times for 5 min each at room temperature and mounted with 250 µL per well of Vectashield Plus Antifade Mounting Medium with DAPI (Vector Laboratories, H-2000).

### Fixation of RanGAP1$^{1-18-WT}$-mEGFP and RanGAP1$^{1-18-4AMUT}$-mEGFP for confocal microscopy

Stable cell lines expressing RanGAP1$^{1-18-WT}$-mEGFP or RanGAP1$^{1-18-4AMUT}$-mEGFP were plated on 4-well Ibidi µ-slides and allowed to attach overnight. Cells were washed once with PBS$^{++}$, incubated in transport buffer (20 mM HEPES pH 7.5, 110 mM KOAc, 2 mM Mg(OAC)$_2$, 0.5 mM EGTA, 2 mM DTT, 1x Halt protease/phosphatase inhibitor (Thermo)) for 4 min at room temperature, and washed twice with PBS$^{++}$ again. Cells were fixed in 3% PFA for 15 min at room temperature, washed three times in TBST, and mounted with 250 µL per well of Vectashield Plus Antifade Mounting Medium with DAPI.

### Preparation of cells expressing 2xGFP-NES-IRES-2xRFP-NLS, NLS-mCherry-LEXY, and NES-mCherry-LiNUS for live imaging

Stable, FACS-sorted wild type RPE1 and RanGAP1$^{\Delta NES}$ clone 1 cells expressing 2xGFP-NES-IRES-2xRFP-NLS were plated on 4-well Ibidi µ-slides and allowed to attach overnight. 50 nM LMB was added for 21–24 h where indicated. Media was supplemented with 1x NucSpot 650 (Biotium) to stain nuclei prior to imaging. Wild type RPE1 cells and RanGAP1$^{\Delta NES}$ clones 1 and 2 stably expressing NLS-mCherry-LEXY or NES-mCherry-LiNUS (but not FACS-sorted) were prepared for imaging in the same way except without LMB addition.

### Imaging and calculation of N/C ratios

Imaging of fixed cells was done on a laser-scanning Leica TCS SP8 confocal microscope with the 63x/1.4 HC PL APO CS2 objective in oil and using the white light laser for excitation (MSK Molecular Cytology Core Facility). Z-stack images in steps of 0.5 µm were acquired and the images with the maximum DAPI signal +/− 1 µm were 2D-projected using average projection in FIJI[82] and fed into custom CellProfiler[83] pipelines. Briefly, for calculation of RanGAP1 N/C ratios, nuclei were first automatically identified from the DAPI signal using adaptive Otsu thresholding and single nuclei were manually confirmed. Next, cell boundaries were automatically identified from the RanGAP1 signal using the "Watershed – Image" method and a global robust background thresholding strategy. Correctly identified cell boundaries were manually confirmed and total cytoplasmic RanGAP1 signal intensity was calculated by subtracting total nuclear RanGAP1 signal intensity from total whole cell RanGAP1 signal intensity. Nuclear ring staining signal was not counted as part of the nuclear signal (i.e., it was assumed to be cytoplasmic). Nuclear to cytoplasmic ratio was calculated by dividing the average RanGAP1 nuclear signal intensity by the average RanGAP1 cytoplasmic signal intensity. N/C ratios for Ran were calculated by using the RanGAP1 signal to mark cell boundaries and dividing average nuclear over the average cytoplasmic Ran signal using a very similar custom CellProfiler pipeline as that used for RanGAP1 calculations but without taking into account the nuclear ring signal, which was not pronounced for Ran. Live imaging of cells expressing 2xGFPNES-IRES-2x-RFPNLS was done via the same microscope setup as for fixed cells but with the addition of the Tokai Hit STXG incubation chamber (MSK Molecular Cytology Core Facility) set to 37 °C and 5% CO$_2$. Z-stack images in 0.5 µm steps were acquired as described above and the images with the maximum NucSpot 650 signal +/− 1 µm were 2D-projected using average projection in FIJI. N/C ratios for cells were calculated by using the GFP signal to mark cell boundaries, NucSpot 650 signal to mark nuclei, and the same custom CellProfiler pipeline used for Ran N/C calculations.

### Imaging and quantification of transport kinetics in NLS-mCherry-LEXY and NES-mCherry-LiNUS cell lines

Cells expressing NLS-mCherry-LEXY or NES-mCherry-LiNUS were grown on 4-well µ-slides and imaged in the presence of 1x NucSpot 650 on the Leica TCS SP8 confocal microscope equipped with the Tokai Hit STXG incubation chamber set to 37 °C and 5% CO$_2$ and the 20x/0.75 HC PL APO CS2 objective immersed in oil. Cells were incubated in the dark for at least 10 min prior to initiation of nuclear export (LEXY) or import (LiNUS) of mCherry, which was induced by 458 nm excitation via the Argon laser at 19.8 W output power (20% of maximum laser power). Pulsatile excitation at 458 nm continued every 2.57 s and images in the mCherry and NucSpot 650 channels were collected concurrently for 5–10 min. Images were processed in FIJI as follows. First, nuclear masks were made using the NucSpot 650 signal and thresholding via the Default method. Next, the Trackmate[84] plugin was used to track individual nuclei and measure their mCherry fluorescence over time. Lastly, mCherry fluorescence was plotted against time and fit to a single exponential in Prism and the half-life of each curve was calculated.

### Flow cytometry and mitotic index determination

Wild type RPE1 cells or RanGAP1$^{\Delta NES}$ clones 1 and 2 were grown on 10 cm plates to 40–70% confluence and pulsed with 10 µM BrdU (BD Pharmigen flow kit) for 45 min. Cells were trypsinized and fixed with 2% PFA in PBS (Biotium) for 10 min at room temperature. Fixed cells were washed with FACS staining buffer (PBS with 3% FBS (Sigma) and 0.03% azide) followed by a wash with ice cold PBS and permeabilized with 100% MeOH (1 mL per 1 × 10$^6$ cells) overnight at −20 °C. Permeabilized cells were washed several times in FACS staining buffer, resuspended in 100 µL DNAse (300 µg/mL, BD Pharmigen flow kit), and incubated

for 1 h at 37 °C. Cells were washed with FACS staining buffer and incubated with α-BrdU antibody conjugated to APC (1:50, BD Pharmigen flow kit) and α-pHH3 (Ser10) antibody conjugated to R-phycoerythrin (2.5 μg/mL, BioLegends) for 20 min at room temperature. Antibody-stained cells were washed in FACS staining buffer and resuspended in 20 μL 7-actinomycin D (7-AAD, BD Pharmigen flow kit) for 5 min at room temperature. 250 μL of FACS staining buffer was added and cells were sorted on a CytoFLEX LX Flow cytometer (Beckman, MSK Flow Cytometry Core Facility) using appropriate lasers for PE (585 nm), 7-AAD (690 nm), and APC (660 nm). Data was processed with FlowJo (BD) and cell populations in G0/G1, S, and G2/M were visualized by plotting the α-BrdU-APC signal vs. 7-AAD whereas mitotic cells were visualized by plotting the α-BrdU-APC signal vs. the α-pHH3-PE signal.

## Western blots

Cell pellets were lysed in 50 mM Tris pH 7.5, 150 mM NaCl, 0.5 mM EDTA, 2 mM DTT, and 2% SDS at 70 °C and passed through an 18 gauge needle several times prior to quantification of total protein via Bradford. 4x LDS (Thermo) and DTT were added to the sample to a final 1x concentration (40 mM DTT in 1x) and 10-30 μg of total protein was loaded onto 4–12% Bis-Tris gels (Thermo) and run in MOPS or MES buffer (Thermo). Gels were transferred via the Trans-Blot Turbo transfer instrument (BioRad) at a constant current of 2.5 A and variable voltage of up to 25 V for 10 min using the Trans-Blot Turbo Mini 0.2 μm PVDF Transfer Packs (BioRad). Membranes were blocked for 2 h at room temperature in Tris-buffered saline (TBS) + 0.1% Tween-20 + 5% non-fat milk (block buffer) and incubated with primary antibodies overnight at 4 °C in TBS + 0.1% Tween-20 + 2% non-fat milk (Ab buffer). Blots were washed in TBS + 0.1% Tween-20 (Western wash buffer) and incubated with secondary antibodies in Ab buffer for 1 h at room temperature. Blots were washed in Western wash buffer, developed using ECL Prime Western Blotting detection reagent (Cytiva), and imaged on a ChemiDoc XRS+ imager (BioRad). The following primary antibodies were used for Western blots: α-RanGAP1 (Sigma, HPA062034); α-β-actin (Santa Cruz, sc81178); α-β-actin (Novus, NBP1-47423); α-Ran (BD, 610340).

## Reporting summary

Further information on research design is available in the Nature Portfolio Reporting Summary linked to this article.

## Data availability

Cryo-EM density maps and atomic coordinates are deposited in the Electron Microscopy Data Bank (EMDB) and Protein Data Bank (PDB) with accession codes 9B62 (RanBP2/Ran(GTP)/SUMO1-RanGAP1/Ubc9/Crm1/Ran(GTP) model); EMDB-44235 [https://www.ebi.ac.uk/emdb/EMD-44235] (RanBP2/Ran(GTP)/SUMO1-RanGAP1/Ubc9/Crm1/Ran(GTP) - composite map); EMDB-44236 [https://www.ebi.ac.uk/emdb/EMD-44236] (RanBP2/SUMO1-RanGAP1/Ran(GTP)/Ubc9/Crm1/Ran(GTP) complex - overall map); EMDB-44237 [https://www.ebi.ac.uk/emdb/EMD-44237] (Crm1/Ran(GTP) – focused refinement map); EMDB-44238 [https://www.ebi.ac.uk/emdb/EMD-44238] (RanBP2 RBD4/Ran(GTP)/RanGAP1 GAP domain – overall map); EMDB-44239 [https://www.ebi.ac.uk/emdb/EMD-44239] (RanBP2 RBD4/Ran(GTP) - focused refinement map); EMDB-44240 [https://www.ebi.ac.uk/emdb/EMD-44240] (RanGAP1 GAP domain – focused refinement map); EMDB-44241 [https://www.ebi.ac.uk/emdb/EMD-44241] (RanBP2/Ran(GTP)/SUMO1-RanGAP1/Ubc9/Crm1/Ran(GTP) complex where SUMO1-RanGAP1/Ubc9 was present – overall map); EMDB-44242 [https://www.ebi.ac.uk/emdb/EMD-44242] (RanGAP1 CTD-SUMO1/Ubc9/RanBP2 E3 domain – focused refinement map); EMDB-44243 [https://www.ebi.ac.uk/emdb/EMD-44243] (RanGAP1 CTD-SUMO1/Ubc9/RanBP2/Crm1/Ran(GTP) – focused refinement map). All stable cell lines and plasmids generated in this study are available from the corresponding author. Any additional information required to reanalyze the data reported in this paper is available from the corresponding author upon request. Source data are provided with this paper.

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

## Acknowledgements

We thank members of the Lima lab for advice and Jason De La Cruz of the MSK Richard Rifkind Center for Cryo-EM funded in part by the NCI Cancer Center Support Grant (CCSG, P30 CA008748) for assistance during cryo-EM data collection. This research was supported in part by an NIH Grant R35 GM118080 (C.D.L.). The content is solely the responsibility of the authors and does not represent the official views of the National Institutes of Health. C.D.L. is an investigator of the Howard Hughes Medical Institute.

## Author contributions

V.B., M.A.D., and C.D.L. designed experiments. M.A.D. and C.D.L. designed substrates for structural work. M.A.D. collected cryo-EM data and M.A.D. and C.D.L. determined structures. V.B. generated cell lines and performed cell assays and biochemical experiments. V.B. and C.D.L. wrote the initial draft and edited the manuscript.

## Competing interests

The authors declare no competing interests.
