## [Peer Review file · Nature Communications]

Structural basis for a nucleoporin exportin complex between RanBP2, SUMO1-RanGAP1, the E2 Ubc9, Crm1 and the Ran GTPase

Corresponding Author: Dr Christopher Lima

Version 0:

Reviewer comments:

Reviewer #1

(Remarks to the Author)

Baytshtok et al. use cryo-EM and complementary biochemical and cellular assays to reveal the structural organization and interactions formed between human RANBP2 (NUP358), CRM1, SUMO1-modified RANGAP1, UBC9, and RAN(GTP). The authors report: (i) interactions formed by two flanking phenylalanine-glycine (FG)-repeat regions supplied by RANBP2 with the convex face of CRM1, including a novel binding site between HEAT repeats 5-7; (ii) structural determination of a RANGAP1•RAN(GTP)•RANBP2RBD4 complex, with recombinant components sourced from a single species, validating the previously reported model that switch-II stabilization is the primary source of RAN GTPase activation upon binding RANGAP1; (iii) identification of a CRM1 nuclear export signal (NES) at the N-terminus of RANGAP1, the deletion of which alters subcellular RANGAP1 and RAN localization to the nucleus and cytoplasm, respectively; (iv) updated RANBP2 E3-ligase activity assays, showing that both interactions with CRM1 and the RAN nucleotide state modulate ubiquitination activity, indicating that the structurally characterized CRM1 complex adopts an inhibited conformation when RAN is locked in the GTP-bound state; and (v) in vivo examination of RANGAP1 NES deletion mutants, which suggest that CRM1-mediated kinetochore recruitment of the UBC9•RANBP2•SUMO1-RANGAP1 complex during mitosis does not require a traditional NES. Overall, this manuscript is a punchy, well-prepared read that will appeal to a broad audience spanning the ubiquitination, nuclear pore structure, and nucleocytoplasmic transport fields. My recommendation is that it should be published in Nature Communications upon completion of minor revisions.

Major comments:

(1) Figure 1. Reconstitution. The SEC experiment is difficult to assess with missing controls. One of the notable strengths of this manuscript is the methods section, a meticulous read containing all the detail a reader could ask for. However, it also makes clear that all the individual proteins and sensible sub-complexes can be purified in isolation. Therefore, a SEC reconstitution experiment lacking control injections for the reagents mixed in isolation is a notable omission. Please repeat this analysis with SEC runs and SDS-PAGE gels for all the independent reagents included.

(2) Figure 1. Selection of the RANBP2 fragment used for cryo-EM. On page 5, the following statements are presented as justification for selecting the RANBP2 fragment containing RBD4 for cryo-EM analysis: "...but size exclusion chromatography (SEC) revealed a broad peak consistent with polydispersity or conformational heterogeneity" and "As RanBP2RBD4 contributed to a more compact architecture." This comparison with larger fragments containing either RBD3 alone or in combination with RBD4 was based on SEC retention volumes and peak widths. However, in the absence of an appropriate standard, these arguments remain speculative, as the relative peak widths appear similar across all three tested samples. Furthermore, inspection of the SDS-PAGE gels shown in Extended Figure S1A suggests that the integration of RAN(GTP) and the similarly sized UBC9 is more evenly distributed throughout the peak fractions in complexes I and II, indicating that a more tightly bound complex is being formed. These reconstitutions should instead be repeated with either SEC-MALS or AUC to provide a quantitative assessment of the overall mass, and thus the relative RAN(GTP) composition.

(3) Figures 5 and 6. Reconstitution of the RANBP2•CRM1•SUMO1-RANGAP1•UBC9•RAN(GTP) complex using RANGAP1 NES deletion mutants. Related to the above discussion, an experiment showing the SEC reconstitution of these complexes with a RANGAP1 NES mutant, such as the one purified for use in Figure 7, would be valuable to determine whether additional contacts made by RANGAP1CTD with CRM1 are sufficient to tether this complex. Thus, offering an explanation for CRM1-dependent recruitment of RANGAP1 lacking the NES to the kinetochore during mitosis.

Minor comments:

- (1) Figure 2. Presentation of the composite RANBP2RBD4•SUMO1-RANGAP1•UBC9•CRM1•RANQ69L(GTP) cryo-EM structure. Panel A, the RANGAP1 NES should be labeled. Panels A and B, when printed the labels for the individual domains are small. Additionally, without reading the extended materials it is unclear which constituents are present stably associated with one another in each of the three overall maps (detailed in Extended Figure S1 G-I), an extra panel displaying these structures, and the relative proportion of particles they are observed within, using the same color and style as the current Panel A would be appreciated.
- (2) Figure 4 and Figure 5. Presentation of the zoom views that highlight contact forming residues. In Figure 4 panels B-E, and Figure 5 panels F-H the semi-transparent colored maps detract from the interpretation of the sidechain contacts, which are the focus of the figure. I would suggest either increasing the transparency of these overlaid maps or omitting them entirely.
- (3) Figure 4. Panel G lacks residue labels.
- (4) Figure 6. Clone labeling for the RANGAP1 NES deletions. Suggest calling these clones delta NES 1 and 2, these are replicates and the jump from clone 2 to clone 10 is unnecessary detail.
- (5) Figure 7. Input gels for the purified reagents used in the E3 ligase activity assay should be added as a separate figure panel or appended to the supplemental source data for Figure 7.
- (6) Extended Figure S4. AlphaFold models in panel E would benefit from labeling or recoloring the N-terminal RANGAP1 NES, so that the relative orientation with the GAP domain was clearer.
- (7) Extended Figure S7. SDS-PAGE gels lack protein labels for each band.
- (8) Extended Table 1. Cryo-EM data collection and refinement statistics. This table should be expanded to provide the following additional detail. Data collection tab – (1) Microscope model, detector/camera modes, automation software. (2) PDB and EMDB codes for the deposited structure. (3) Was an energy filter applied? If so, what was the slit width (Å²). (4) Exposure time, number of frames, micrographs collected/used. (5) Sphericity, at FSC 0.5. Refinement tab – (1) CCvol/mask. (2) EMRinger score. (3) CaBLAM outliers.

Reviewer #2

(Remarks to the Author)

RanBP2 is a multi-functional nucleoporin (Nup358) consists various functional motifs and has broad range of biochemical/cell biological functions. Obvious from its name, RanBP2 has Ran small GTPase binding motif (RanBD) that interact GTP bound form of Ran preferably and contains FG repeats that interact karyopherin and is frequently observed in nucleoporins. These motifs have key role in nucleo/cytoplasmic transport driven by RanGTPase and karyopherin. In addition to that role, RanBP2 interact to SUMOylated form of RanGap1 and Ubc9, SUMO conjugating enzyme (SUMO-E2) then has biochemical activity as SUMO ligase. Extensive works from the corresponding authors lab and others demonstrated domains required for SUMO ligase activity and it is determined that multiple functional motifs, including FG repeats and RanBDs, are located proximal to motif required for SUMO ligase. Thus, it has been a major question how these distinct molecule interactions contribute SUMO ligase activity of RanBP2 in physiological context. Previous report indicated Interaction to Ubc9 and SUMO moiety plays a critical role for SUMO ligase function of RanBP2 and discharging SUMO moiety from Ubc9 by transmitting SUMO to substrate plays critical role on SUMO ligase activity. However, it is also reported that SUMO1 modified form of RanGAP1 bound to RanBP2 at the domains overlapping SUMO ligase domain with high affinity to be retained as a stable complex. This observation questions how UBC9-SUMO2/3 adduct can coordinated with SUMO1-RanGAP1/RanBP2/Ubc9 complex to act as SUMO ligase. In addition, Crm1 (as known as exprotin 1) can form stable interaction to RanBP2 during mitosis that promotes kinetochore localization of RanBP2 complex, which is expected to contribute mitotic centromeric SUMOylation. Thus, FG repeat contribution to Crm1/RanBP2 complex and its role in E3 ligase activity provide additional layer of potential regulatory mechanism of RanBP2's SUMO ligase activity.

This manuscript addressed these challenging questions by solving structure of the complex by CryoEM. The obtained 3D structural information determined mode of interaction between RanBP2 and each binding partner, and special relationship among all components, RanBP2, RanGTP, SUMO1-RanGAP1, Ubc9 and Crm1. The structure demonstrated novel interaction between Crm1 and RanGAP1, which both C-terminal domain of RanGAP1 and N-terminal NES sequence are contributed. Using well-established in vitro SUMOylation assay, contribution of this complex formation with the context with RanGTP cycle were examined and the result suggested RanGTP could negatively affect SUMO ligase activity of RanBP2 for SUMO2 conjugation of Borealin (a model substrate for mitotic centromeric SUMOylation). RanGAP1-NES/Crm1 interaction, which can be stabilized by RanGTP, reduces SUMO ligase activity toward Borealin, suggesting potential mechanism of inhibitory effect of RanGTP on SUMO ligase activity.

Overall, this challenging structural analysis on RanBP2 complex provided intriguing potential regulatory mechanism of SUMO ligase activity of RanBP2. This potential mechanism can be tested in cellular context, as such mitotic SUMOylation with ΔNES-RanGAP1 or effect of leptomycin B on SUMO ligase activity and provided insight of how mitotic RanGTP gradient at vicinity of chromosome and kinetochore localization of RanBP2 complex can act on mitotic chromosomal SUMOylation. With the highly informative mechanistic insight of this structural analysis and expected contribution in broad area of cell biology, as such mitosis, SUMOylation, genome stability, & Nucleo/Cytoplasmic transport, I support publishing this manuscript in this journal.

Some suggestion of minor (or quick experiment) for improvement are below.

- 1) The stabilized complex formation with Crm1/RanGAP1-NES interaction on SUMO ligase activity is intriguing model. This imply the presence of RanGTP might reduce SUMO ligase function. That sound contradictory if the complex located vicinity of mitotic chromosome where RanGTP is higher concentration by chromosomal RCC1. Overall, additional discussion including possible cellular context could help provoking the future investigation at cellular level.
- 2) Related to above issue, is it possible to test effect of leptomycin B with in vitro Borealin SUMOylation assay? This might

be just confirmation of NES/Crm1 interaction but could support the importance of Crm1/RanGAP1-NES interaction model on SUMO ligase activity, if leptomycin B can inhibit the SUMO ligase activity with full length RanGAP1 containing complex but not affect Δ NES-RanGAP1 containing complex.

3) If possible, indicating Cysteine residue of Ubc9 in the complex structure might help better understanding how Ubc9 work in this complex. That might help to see if the Ubc9 within the complex can be accessible for E1-SUMO adduct to transfer SUMO moiety. If not, then IR2-mediated additional Ubc9-SUMO2 interaction model could be supported and Crm1 contribution on that makes more sense.

4) For Δ NES-RanGAP1 cell line, it might be helpful if the kinetochore RanBP2 localization in mitosis can be analyzed more detail, because Crm1/RanBP2 interaction is shown to have key role in kinetochore localization of RaBP2 complex. Thus, that analysis could add meaningful information on RanGAP1-NES/Crm1 interaction in cellular context.

Reviewer #3

(Remarks to the Author)

Review Manuscript NSMB 155416_0

In the research manuscript entitled "Structural basis for a nucleoporin exportin complex between RanBP2, SUMO1-RanGAP1, the E2 Ubc9, Crm1 and the Ran GTPase" Vladimir Baytshtok, Michael A. DiMattia and Christopher D. Lima present the cryo-EM structure (3,18 Å overall) of the human complex of the proteins mentioned in the title encompassing 1 molecule of the proteins, except for 2 molecules of RanGTP, in a state most likely found at the cytosolic side of the NPC.

Moreover, they demonstrate biochemical, biophysical and cell-based results investigating individual aspects, like the newly identified NES of RanGAP by comparing wild type and mutant forms of RanGAP-NES or the E3 ligase activity of RanBP2. Moreover, the authors identify regions essential for direct interactions and investigate cellular localization and function.

Overall, this is a detailed and ground breaking work to understand the modes of interaction of the different binding partners to RanGAP, which clearly warrants publication in Nature Communication. The few points below under major/medium should be addressed. This could improve understanding for a more general readership.

Major point:

The discussion reads more like a synopsis of the results presented. Most of the results are presented and concluded within the scientific context in the results section, but some results are left uninterpreted (eg. p12). As these interpretations seems to be more speculative, they could be discussed here, like the conclusions and reasoning for the results observed in the RanGAP Δ NES Ran GTPase-distribution and transport experiments (Figs 5, 6).

Medium point

In figure 6 according to the staining, the amount of Ran seems increased in mutant cells, which in turn could explain the observed effects. The Western blot results indicates otherwise. A quantification of the cells, or the quantification results of the blot or a blot with the actin control at identical level would be helpful. In the "+ LMB" experiment, the wt cells are larger than in the other panels. Is this really the case? Please comment.

Minor points:

p5: 17 kDa loss and shift of peak: could this be caused by inter-complex binding due to the presence of 2 binding domains?

p6: the presence of 2 RanGTPs in the complex is not clearly stated. Perhaps mention 2 RanGTPs under title: overall architecture, title p 6: Crm1 interacts with second RanGTP?

p8 : Based on the *S. pombe* RanGAP1-Ran(GTP)-RanBP1 structure, it...

p9: motifs from protein kinase inhibitor (PKI) and Snurportin1 (Spn1; Extended Data Fig. 4D).

Naming

In the original Literature RCC1 and SPN1 are used versus Rcc1 and Spn1 and Spn (p. 3, 14, 15, 50) are used in text.

Typos

- In vitro not in italic, p14 p59

- pET-28a not pet28a or otherwise (not identically used within the text, e.g. p. 50)

- L not I for liter as in some cases (e.g. p. 53 bottom, p. 54 top, 55, 56)

- RANGAP instead RanGAP p.61

- p. 66 After 5-7 days, cells were moved into the same DME:F12 media but supplemented with 300 μ g/mL G418 (Sigma) instead of 1% pen/strep. => 10gram/L??

Reviewer #4

(Remarks to the Author)

The nucleoporin RanBP2/Nup358 is a SUMO E3 ligase localized at the nuclear pore complex and promotes the disassembly of Crm1-dependent export complexes. Baytshtok et al determined a high resolution structure of a RanBP2/SUMO1-RanGAP1/Ubc9/Crm1/Ran(GTP) complex, providing the most holistic view of this nucleoporin exportin complex to date.

The structure uncovers previously unknown interactions between Crm1 and RanGAP1. An N-terminal helix of RanGAP1 binds to the Crm1 NES-binding pocket and tethers the RanGAP1-GAP/Ran/RanBP2-RBD4 module to Crm1. The authors demonstrate that this N-terminal helix functions as an NES for RanGAP1 in cells. In addition, the structure shows that the SUMO1-RanGAP1-CTD/Ubc9/RanBP2-IR1 module directly interacts with Crm1.

The structure also reveals the structural basis for RanBP2 FG repeat interactions with Crm1. Furthermore, the model of the RanGAP1-GAP/Ran/RanBP2-RBD4 module provides insights into Ran activation by RanGAP1.

The cryo-EM reconstruction is of excellent quality, and the model is well justified. Using an in vitro SUMO conjugation assay, the authors demonstrate that E3 ligase activity depends on RanBP2-IR1 and is modulated by Crm1, RanGAP1-NES, and the Ran GTPase cycle. These findings corroborate with the structural findings.

Overall, this is an impressive work and greatly expands our understanding of the mechanism of RanBP2 in nuclear export and SUMO-related functions. This manuscript will make an exceptional addition to the journal. I only have a few minor points as listed below.

1. Fig 1A: It would be helpful to define some of the domain abbreviations (such as OE, RBD) in the legend.
2. Comparison with PDB 1K5D (S. pombe RanGAP1) on pg 8 lacks context. Perhaps a supplemental figure can be added.
3. Catalytic glutamine of Ran (pg 8): The residue number should be pointed out in the text.
4. EMD ID should be added in Extended Data Figures 1 and 2, and Supplementary Table 1 as applicable. Currently, it is hard to correlate the various maps with the EMD entries.

Reviewer #5

(Remarks to the Author)

Version 1:

Reviewer comments:

Reviewer #1

(Remarks to the Author)

Baytshtok and colleagues present a compelling manuscript addressing the structural and functional characterization of a challenging and conformationally dynamic complex comprising the human nuclear export factor CRM1•Ran(GTP) with the cytoplasmic filament nucleoporin constituent RanBP2 (NUP358)•SUMO1-RanGAP1•UBC9, which localizes either at the NPC or, during mitosis is recruited to the Kinetochore. Overall, the revisions implemented in response to both my feedback and that of the other reviewers has yielded a polished manuscript that markedly enhances the rigor of the experimental SEC reconstitutions and the clarity of the graphical presentations compared to the previous submission. I particularly appreciated the comprehensive reconstitution experiments with appropriate individual protein controls, presented in Figure S1; and the refined graphical layout in Figure 2A, which clarify the cryo-EM workflow and succinctly highlights regions exhibiting conformational heterogeneity. As all my comments have been addressed, I reiterate my strong support for the timely publication of this punchy and meticulously prepared manuscript in Nature Communications without further revision.

Reviewer #2

(Remarks to the Author)

This revised manuscript (NCOMMS-25-06168A) addressed most of my concerns on previous manuscript. I appreciate that they provided additional data for in vitro SUMOylation assay with Leptomycin B and performed SEC reconstitution assay with additional controls. Discussion section is improved by addressing multiple comments from reviewers.

Overall, I truly appreciate the authors' efforts on addressing reviewers' concerns to strengthen the conclusion of this study. I fully support publishing this manuscript in the Nature communications.

Reviewer #3

(Remarks to the Author)

The suggestions made have been fully implemented in the revised version of the manuscript and I have no further comments. I strongly recommend publication in Nature communications.

We thank the reviewers for their insightful comments and suggestions and for their positive reviews of the manuscript. In response to reviewer comments, we have made the following major changes to the manuscript:

- 1) We performed size exclusion chromatography experiments with SDS-PAGE analyses to generate a new Supplementary Fig.1. We re-numbered the other supplementary figures accordingly.**
- 2) We added 3 new panels in the main figures (1C, 2A, and 6G), 2 new panels in supplementary figures (7C and 8C), and 1 new panel in source data for Fig. 7 (panel C) with all panels except Fig. 2A generated from new or newly replicated experiments.**
- 3) We reorganized Supplementary Fig. 2 A and B (formerly Supplementary Fig.1) to only show the full chromatograph and gel fractions for the complex that was used for cryo-EM.**
- 4) We revised Fig. 4G to more clearly illustrate the would-be clash between the RanGAP1 in our structure and the Ran(GMPPNP) in 7MNZ.**
- 5) An updated Supplementary Table 1 is provided detailing cryo-EM data collection and analysis.**
- 6) EMDB identifiers were added to Supplementary Figures 2 and 3 (formerly Supp Figs 1 and 2).**
- 7) EM maps in Figs. 4(B-E) and 5(G,H) were made more transparent**
- 8) New labels were added in Supp Fig. 4, Supp. Fig. 5E, Supp. Fig. 8, and Source Data for Fig. 7.**
- 9) Various typos were fixed, format of several protein and vector names has been updated (e.g. RCC1, SPN1, pET-28b), and Δ NES clones 2 and 10 have been renamed clones 1 and 2.**
- 10) The Discussion section was extended to include further analysis of localization and transport assay results as well as implications of our biochemical assays for SUMO E3 ligase activity of RanBP2 during mitosis.**
- 11) The Methods section was updated to include a description of analytical gel filtration runs in Supplementary Fig. 1 and reaction conditions for Supplementary Fig. 8C. Methods subsection describing purification of Snurportin 1 was removed as this was erroneously included from a previous version of the manuscript.**

The following point-by-point responses outline our changes to the manuscript as per reviewers' critiques.

Reviewer #1 (Remarks to the Author):

Baytshtok et al. use cryo-EM and complementary biochemical and cellular assays to reveal the structural organization and interactions formed between human RANBP2 (NUP358), CRM1, SUMO1-modified RANGAP1, UBC9, and RAN(GTP). The authors report: (i) interactions formed by two flanking phenylalanine-glycine (FG)-repeat regions supplied by RANBP2 with the convex face of CRM1, including a novel binding site between HEAT repeats 5-7; (ii) structural determination of a RANGAP1•RAN(GTP)•RANBP2RBD4 complex, with recombinant components sourced

from a single species, validating the previously reported model that switch-II stabilization is the primary source of RAN GTPase activation upon binding RANGAP1; (iii) identification of a CRM1 nuclear export signal (NES) at the N-terminus of RANGAP1, the deletion of which alters subcellular RANGAP1 and RAN localization to the nucleus and cytoplasm, respectively; (iv) updated RANBP2 E3-ligase activity assays, showing that both interactions with CRM1 and the RAN nucleotide state modulate ubiquitination activity, indicating that the structurally characterized CRM1 complex adopts an inhibited conformation when RAN is locked in the GTP-bound state; and (v) in vivo examination of RANGAP1 NES deletion mutants, which suggest that CRM1-mediated kinetochore recruitment of the UBC9•RANBP2•SUMO1-RANGAP1 complex during mitosis does not require a traditional NES. Overall, this manuscript is a punchy, well-prepared read that will appeal to a broad audience spanning the ubiquitination, nuclear pore structure, and nucleocytoplasmic transport fields. My recommendation is that it should be published in Nature Communications upon completion of minor revisions.

Major comments:

(1) Figure 1. Reconstitution. The SEC experiment is difficult to assess with missing controls. One of the notable strengths of this manuscript is the methods section, a meticulous read containing all the detail a reader could ask for. However, it also makes clear that all the individual proteins and sensible sub-complexes can be purified in isolation. Therefore, a SEC reconstitution experiment lacking control injections for the reagents mixed in isolation is a notable omission. Please repeat this analysis with SEC runs and SDS-PAGE gels for all the independent reagents included.

This is an important point that we overlooked, and we thank the reviewer for this suggestion. We performed new SEC reconstitution experiments with appropriate control injections and their accompanying gels, which can be found in the new Supplementary Fig. 1.

(2) Figure 1. Selection of the RANBP2 fragment used for cryo-EM. On page 5, the following statements are presented as justification for selecting the RANBP2 fragment containing RBD4 for cryo-EM analysis: "...but size exclusion chromatography (SEC) revealed a broad peak consistent with polydispersity or conformational heterogeneity" and "As RanBP2RBD4 contributed to a more compact architecture." This comparison with larger fragments containing either RBD3 alone or in combination with RBD4 was based on SEC retention volumes and peak widths. However, in the absence of an appropriate standard, these arguments remain speculative, as the relative peak widths appear similar across all three tested samples. Furthermore, inspection of the SDS-PAGE gels shown in Extended Figure S1A suggests that the integration of RAN(GTP) and the similarly sized UBC9 is more evenly distributed throughout the peak fractions in complexes I and II, indicating that a more tightly bound complex is being formed. These reconstitutions should instead be repeated with either SEC-MALS or AUC to provide a

quantitative assessment of the overall mass, and thus the relative RAN(GTP) composition.

We agree that the “compact architecture” statement was speculative and have changed the text to reflect this uncertainty (p. 5 – “Additionally, as both RanBP2^{RBD3} and RanBP2^{RBD4} fragments are ~70 kDa and contain a single RBD, complexes containing these variants would be expected to have similar Stokes radii and SEC elution profiles, further suggesting that the different elution profile observed for the RanBP2^{RBD4} complex may be a result of a more compact architecture or less conformational heterogeneity.” Unfortunately, we do not currently have easy access to AUC or SEC-MALS to definitively test differences in the relative Ran(GTP) and/or Ubc9 compositions of the three complexes. However, our new SEC analyses (see Supp Fig. 1(A-C)) suggest that all RanBP2^{RBD4} is bound to Ran(GTP) and SUMO1-RanGAP1/Ubc9 with no free RanBP2^{RBD4} or SUMO1-RanGAP1/Ubc9 migrating separately (Ran is used in excess in panel C). This of course does not exclude the possibility that complex III is less stable than the other two complexes with respect to Ran(GTP), but we feel that this is currently beyond the scope of this manuscript. While not included in this manuscript, our attempts at cryo-EM analysis of complexes containing the RanBP2^{RBD3-4} fragment met with limited success presumably due to the conformational heterogeneity of the sample in comparison to complexes containing the RanBP2^{RBD4} variant.

(3) Figures 5 and 6. Reconstitution of the RANBP2•CRM1•SUMO1-RANGAP1•UBC9•RAN(GTP) complex using RANGAP1 NES deletion mutants. Related to the above discussion, an experiment showing the SEC reconstitution of these complexes with a RANGAP1 NES mutant, such as the one purified for use in Figure 7, would be valuable to determine whether additional contacts made by RANGAP1CTD with CRM1 are sufficient to tether this complex. Thus, offering an explanation for CRM1-dependent recruitment of RANGAP1 lacking the NES to the kinetochore during mitosis.

We thank the reviewer for this suggestion and have now included several SEC reconstitution experiments with the SUMO1-RanGAP1^{ΔN18} NES deletion mutant in complex with Ubc9, RanBP2^{RBD4}, Crm1, and Ran^{Q69L} in Supplementary Fig. 1 (D-F). Additionally, as has been shown previously (see Ritterhoff et al., *Nat Comm*, 2016, especially Supplementary Figure 1), Crm1 can still bind RanBP2 in the absence of SUMO1-RanGAP1/Ubc9 as well as in the presence of a SUMO1-RanGAP1 variant that lacks the N-terminal catalytic GAP domain, results consistent with NES-independent recruitment of RanBP2/SUMO1-RanGAP1/Ubc9 by Crm1 in mitosis. We now explicitly state that the RanGAP1 NES is not required for association of RanBP2 complexes with Crm1 at the top of p. 14 of the revised manuscript.

Minor comments:

(1) Figure 2. Presentation of the composite RANBP2RBD4•SUMO1-RANGAP1•UBC9•CRM1•RANQ69L(GTP) cryo-EM structure. Panel A, the RANGAP1 NES should be labeled. Panels A and B, when printed the labels for the individual domains are small. Additionally, without reading the extended materials it is unclear which constituents are present stably associated with one another in each of the three overall maps (detailed in Extended Figure S1 G-I), an extra panel displaying these structures, and the relative proportion of particles they are observed within, using the same color and style as the current Panel A would be appreciated.

We note that all constituents of the complex (RanBP2^{RBD4}, Crm1, Ran^{Q69L}, and SUMO1-RanGAP1/Ubc9) appear stably associated in all particles where Crm1 is bound to Ran^{Q69L}, as exemplified by the presence of RanBP2 FG repeats and the RanGAP1 NES in all these particles. Therefore, our composite map is not a composite of different subcomplexes but rather a composite of different conformational states of the whole complex. We attempt to illustrate this more clearly through a new panel A in Figure 2 of the revised manuscript where the overall maps are directly correlated with the alternative conformations assumed by the various components of our complex using the same color scheme as in panels B and C, as requested by the reviewer.

(2) Figure 4 and Figure 5. Presentation of the zoom views that highlight contact forming residues. In Figure 4 panels B-E, and Figure 5 panels F-H the semi-transparent colored maps detract from the interpretation of the sidechain contacts, which are the focus of the figure. I would suggest either increasing the transparency of these overlaid maps or omitting them entirely.

We addressed these concerns by increasing the transparency of the overlaid maps.

(3) Figure 4. Panel G lacks residue labels.

We revised this panel to show a different view and have added the residue labels for residues that would clash between RanGAP1 and Ran from PDB 7MNZ. We represent Ran^{Q69L}(GTP) only in cartoon form and omit residue labels for clarity.

(4) Figure 6. Clone labeling for the RANGAP1 NES deletions. Suggest calling these clones delta NES 1 and 2, these are replicates and the jump from clone 2 to clone 10 is unnecessary detail

We have renamed clones 2 and 10 as clones 1 and 2 as suggested.

(5) Figure 7. Input gels for the purified reagents used in the E3 ligase activity assay should be added as a separate figure panel or appended to the supplemental source data for Figure 7.

We appended a gel of the purified reagents used in E3 ligase activity assays in Source Data for Figure 7, panel C.

(6) Extended Figure S4. Alphafold models in panel E would benefit from labeling or recoloring the N-terminal RANGAP1 NES, so that the relative orientation with the GAP domain was clearer.

We labelled the RanGAP1 NES in the new Supplementary Fig. 5

(7) Extended Figure S7. SDS-PAGE gels lack protein labels for each band.

We added protein labels for each band on the Coomassie gels in the new Supplementary Fig. 8.

(8) Extended Table 1. Cryo-EM data collection and refinement statistics. This table should be expanded to provide the following additional detail. Data collection tab – (1) Microscope model, detector/camera modes, automation software. (2) PDB and EMDB codes for the deposited structure. (3) Was an energy filter applied? If so, what was the slit width (Å²). (4) Exposure time, number of frames, micrographs collected/used. (5) Sphericity, at FSC 0.5. Refinement tab – (1) CCvol/mask. (2) EMRinger score. (3) CaBLAM outliers.

The requested information is now supplied in updated Supplementary Table 1.

Reviewer #2 (Remarks to the Author):

RanBP2 is a multi-functional nucleoporin (Nup358) consists various functional motifs and has broad range of biochemical/cell biological functions. Obvious from its name, RanBP2 has Ran small GTPase binding motif (RanBD) that interact GTP bound form of Ran preferably and contains FG repeats that interact karyopherin and is frequently observed in nucleoporins. These motifs have key role in nucleo/cytoplasmic transport driven by RanGTPase and karyopherin. In addition to that role, RanBP2 interact to SUMOylated form of RanGap1 and Ubc9, SUMO conjugating enzyme (SUMO-E2) then has biochemical activity as SUMO ligase. Extensive works from the corresponding authors lab and others demonstrated domains required for SUMO ligase activity and it is determined that multiple functional motifs, including FG repeats and RanBDs, are located proximal to motif required for SUMO ligase. Thus, it has been a major question how these distinct molecule interactions contribute SUMO ligase activity of RanBP2 in physiological context. Previous report indicated Interaction to Ubc9 and SUMO moiety plays a critical role for SUMO ligase function of RanBP2 and discharging SUMO moiety

from Ubc9 by transmitting SUMO to substrate plays critical role on SUMO ligase activity. However, it is also reported that SUMO1 modified form of RanGAP1 bound to RanBP2 at the domains overlapping SUMO ligase domain with high affinity to be retained as a stable complex. This observation questions how UBC9-SUMO2/3 adduct can coordinated with SUMO1-RanGAP1/RanBP2/Ubc9 complex to act as SUMO ligase. In addition, Crm1 (as known as exprotin 1) can form stable interaction to RanBP2 during mitosis that promotes kinetochore localization of RanBP2 complex, which is expected to contribute mitotic centromeric SUMOylation. Thus, FG repeat contribution to Crm1/RanBP2 complex and its role in E3 ligase activity provide additional layer of potential regulatory mechanism of RanBP2's SUMO ligase activity.

This manuscript addressed these challenging questions by solving structure of the complex by CryoEM. The obtained 3D structural information determined mode of interaction between RanBP2 and each binding partner, and special relationship among all components, RanBP2, RanGTP, SUMO1-RanGAP1, Ubc9 and Crm1. The structure demonstrated novel interaction between Crm1 and RanGAP1, which both C-terminal domain of RanGAP1 and N-terminal NES sequence are contributed. Using well-established in vitro SUMOylation assay, contribution of this complex formation with the context with RanGTP cycle were examined and the result suggested RanGTP could negatively affect SUMO ligase activity of RanBP2 for SUMO2 conjugation of Borealin (a model substrate for mitotic centromeric SUMOylation). RanGAP1-NES/Crm1 interaction, which can be stabilized by RanGTP, reduces SUMO ligase activity toward Borealin, suggesting potential mechanism of inhibitory effect of RanGTP on SUMO ligase activity.

Overall, this challenging structural analysis on RanBP2 complex provided intriguing potential regulatory mechanism of SUMO ligase activity of RanBP2. This potential mechanism can be tested in cellular context, as such mitotic SUMOylation with Δ NES-RanGAP1 or effect of leptomycin B on SUMO ligase activity and provided insight of how mitotic RanGTP gradient at vicinity of chromosome and kinetochore localization of RanBP2 complex can act on mitotic chromosomal SUMOylation. With the highly informative mechanistic insight of this structural analysis and expected contribution in bread area of cell biology, as such mitosis, SUMOylation, genome stability, & Nucleo/Cytoplasmic transport, I support publishing this manuscript in this journal.

Some suggestion of minor (or quick experiment) for improvement are below.

- 1) The stabilized complex formation with Crm1/RanGAP1-NES interaction on SUMO ligase activity is intriguing model. This imply the presence of RanGTP might reduce SUMO ligase function. That sound contradictory if the complex located vicinity of mitotic chromosome where RanGTP is higher concentration by chromosomal RCC1. Overall, additional discussion including possible cellular context could help provoking the future investigation at cellular level.

We have extended the Discussion section to include the following (p. 18):

“As Ran(GTP) levels are higher proximal to mitotic chromosomes, it is possible that the RanBP2/SUMO1-RanGAP1/Ubc9/Crm1 complex found at kinetochores has lower E3 ligase activity and that most mitotic SUMOylation by RanBP2 does not occur while it is localized to kinetochores but rather in the mitotic cytosol as has been suggested for topoisomerase II α , for example. Alternatively, RanBP2 SUMO E3 ligase activity may persist at kinetochores because the GAP activity of RanGAP1 would promote Ran GTP-GDP cycling (Fig. 7D-E; Supplementary Fig. 8A-B). Further work is needed to understand the relationship between mitotic localization and SUMO E3 ligase activity of the RanBP2/SUMO1-RanGAP1/Ubc9 complex.”

2) Related to above issue, is it possible to test effect of leptomycin B with in vitro Borealin SUMOylation assay? This might be just confirmation of NES/Crm1 interaction but could support the importance of Crm1/RanGAP1-NES interaction model on SUMO ligase activity, if leptomycin B can inhibit the SUMO ligase activity with full length RanGAP1 containing complex but not affect Δ NES-RanGAP1 containing complex.

We thank the reviewer for this valuable suggestion – we performed a variation of the suggested experiment and show this new data in Supplementary Fig. 8C. Our results show that for complexes with wild type SUMO1-RanGAP1, Crm1 inhibits SUMOylation in the absence of leptomycin B (Fig. 6F – right panel) but that inhibition is relieved in the presence of leptomycin B to levels observed previously for SUMO1-RanGAP1 ^{Δ N18} complexes (Fig. 6F – left panel). These results reinforce the idea that RanGAP1^{NES}-Crm1 interactions attenuate SUMO E3 ligase activity of RanBP2 in the presence of Ran^{Q69L}(GTP).

3) If possible, indicating Cysteine residue of Ubc9 in the complex structure might help better understanding how Ubc9 work in this complex. That might help to see if the Ubc9 within the complex can be accessible for E1-SUMO adduct to transfer SUMO moiety. If not, then IR2-mediated additional Ubc9-SUMO2 interaction model could be supported and Crm1 contribution on that makes more sense

Previous work shows that the Ubc9 molecule in RanBP2/SUMO1-RanGAP1/Ubc9 is noncatalytic and plays a purely structural role (Werner et al. *Mol Cell*, 2012). Experiments from that paper illustrate that another equivalent of catalytically active Ubc9 must be added to RanBP2/SUMO1-RanGAP1/Ubc9 for SUMO conjugation to occur. However, as requested, we labelled the catalytic cysteine of this catalytically-inactive Ubc9 (Cys93) in Supplementary Fig.4 panel 3 as well as the crosslinked lysine 524 of RanGAP1.

4) For Δ NES-RanGAP1 cell line, it might be helpful if the kinetochore RanBP2 localization in mitosis can be analyzed more detail, because Crm1/RanBP2 interaction is shown to have key role in kinetochore localization of RaBP2 complex. Thus, that

analysis could add meaningful information on RanGAP1-NES/Crm1 interaction in cellular context.

At the suggestion of the reviewer, we looked at RanBP2 localization in mitosis by confocal microscopy in wild type and Δ NES RanGAP1 cell lines and present our results in the new Supplementary Fig. 7C. We observe no differences in localization between the cell lines. Additionally, Crm1 binds to the RanBP2^{RBD4}/SUMO1- Δ NES RanGAP1/Ubc9/Ran^{Q69L} complex in vitro, as we now show by SEC in Supplementary Fig. 1 D-F. It has been shown before that RanBP2 binds Crm1 on its own as well as in complex with a RanGAP1 variant lacking the N-terminal GAP domain (Ritterhoff et al., *Nat Comm*, 2016), also supporting the idea that Crm1 recruits RanBP2 and its binding partners to kinetochores in a manner independent of the RanGAP1 NES. We point out the fact that the NES of RanGAP1 is not required for RanBP2-Crm1 interaction explicitly at the top of p. 14 of the revised manuscript.

Reviewer #3 (Remarks to the Author):

Review Manuscript NSMB 155416_0

In the research manuscript entitled “Structural basis for a nucleoporin exportin complex between RanBP2, SUMO1-RanGAP1, the E2 Ubc9, Crm1 and the Ran GTPase” Vladimir Baytshtok, Michael A. DiMattia and Christopher D. Lima present the cryo-EM structure (3,18 Å overall) of the human complex of the proteins mentioned in the title encompassing 1 molecule of the proteins, except for 2 molecules of RanGTP, in a state most likely found at the cytosolic side of the NPC.

Moreover, they demonstrate biochemical, biophysical and cell-based results investigating individual aspects, like the newly identified NES of RanGAP by comparing wild type and mutant forms of RanGAP-NES or the E3 ligase activity of RanBP2. Moreover, the authors identify regions essential for direct interactions and investigate cellular localization and function.

Overall, this is a detailed and ground breaking work to understand the modes of interaction of the different binding partners to RanGAP, which clearly warrants publication in Nature Communication. The few points below under major/medium should be addressed. This could improve understanding for a more general readership.

We thank the reviewer for this synopsis and support.

Major point:

The discussion reads more like a synopsis of the results presented. Most of the results are presented and concluded within the scientific context in the results section, but some results are left uninterpreted (eg. p12). As these interpretations seems to be more speculative, they could be discussed here, like the conclusions and reasoning for the results observed in the RanGAPdeltaNES Ran GTPase-distribution and transport experiments (Figs 5, 6).

We added our interpretations of the mislocalization and transport experiments to the Discussion section (p.17).

Medium point

In figure 6 according to the staining, the amount of Ran seems increased in mutant cells, which in turn could explain the observed effects. The Western blot results indicates otherwise. A quantification of the cells, or the quantification results of the blot or a blot with the actin control at identical level would be helpful. In the "+ LMB" experiment, the wt cells are larger than in the other panels. Is this really the case? Please comment.

We agree with the reviewer that the different actin levels in our prior anti-Ran Western blot complicated interpretation. We repeated the anti-Ran Western blot, which shows that Ran levels between wild type and Δ NES clones are similar, and included this new western in updated Figure 6 panel G. In terms of nuclear size, we repeatedly observe increase in nuclear size upon treatment with leptomycin B, an observation also made in fission yeast (see Neumann and Nurse, *JCB* 2007).

Minor points:

p5: 17 kDa loss and shift of peak: could this be caused by inter-complex binding due to the presence of 2 binding domains?

Since the RBD3 variant elutes similarly to the RanBP2 RBD3-RBD4 variant but still has only one Ran binding domain, it is unlikely that the elution profile shift is due to the presence of two RBDs vs. one. To expand upon this, we inserted the following sentence: "Additionally, as both RanBP2^{RBD3} and RanBP2^{RBD4} fragments are ~70 kDa and contain a single RBD, their complexes would be expected to have similar Stokes radii and SEC elution profiles ..." (p. 5).

p6: the presence of 2 RanGTPs in the complex is not clearly stated. Perhaps mention 2 RanGTPs under title: overall architecture, title p 6: Crm1 interacts with second RanGTP?

We modified the sentence under “Overall architecture” to include 2xRan^{Q69L}(GTP) as suggested by the reviewer. Crm1 interacts with one Ran^{Q69L}(GTP) molecule and RanBP2^{RBD4} and RanGAP1^{GAP} bind the other Ran^{Q69L}(GTP).

p8 : Based on the *S. pombe* RanGAP1-Ran(GTP)-RanBP1 structure, it....

This change was made in the revised manuscript.

p9: motifs from protein kinase inhibitor (PKI) and Snurportin1 (Spn1; Extended Data Fig. 4D).

This change was made in the revised manuscript.

Naming

In the original Literature RCC1 and SPN1 are used versus Rcc1 and Spn1 and Spn (p. 3, 14, 15, 50) are used in text.

We changed the nomenclature as suggested in the revised manuscript.

Typos

- In vitro not in italic, p14 p59
- pET-28a not pet28a or otherwise (not identically used within the text, e.g. p. 50)
- L not l for liter as in some cases (e.g. p. 53 bottom, p. 54 top, 55, 56)
- RANGAP instead RanGAP p.61

All the above have been addressed in the revision.

- p. 66 After 5-7 days, cells were moved into the same DME:F12 media but supplemented with 300 µg/mL G418 (Sigma) instead of 1% pen/strep. => 10gram/L??

We corrected this to “1% pen/strep (vol/vol)”

Reviewer #4 (Remarks to the Author):

The nucleoporin RanBP2/Nup358 is a SUMO E3 ligase localized at the nuclear pore complex and promotes the disassembly of Crm1-dependent export complexes. Baytshtok et al determined a high resolution structure of a RanBP2/SUMO1-RanGAP1/Ubc9/Crm1/Ran(GTP) complex, providing the most holistic view of this nucleoporin exportin complex to date.

The structure uncovers previously unknown interactions between Crm1 and RanGAP1. An N-terminal helix of RanGAP1 binds to the Crm1 NES-binding pocket and tethers the RanGAP1-GAP/Ran/RanBP2-RBD4 module to Crm1. The authors demonstrate that this N-terminal helix functions as an NES for RanGAP1 in cells. In addition, the structure shows that the SUMO1-RanGAP1-CTD/Ubc9/RanBP2-IR1 module directly interacts with Crm1.

The structure also reveals the structural basis for RanBP2 FG repeat interactions with Crm1. Furthermore, the model of the RanGAP1-GAP/Ran/RanBP2-RBD4 module provides insights into Ran activation by RanGAP1.

The cryo-EM reconstruction is of excellent quality, and the model is well justified. Using an in vitro SUMO conjugation assay, the authors demonstrate that E3 ligase activity depends on RanBP2-IR1 and is modulated by Crm1, RanGAP1-NES, and the Ran GTPase cycle. These findings corroborate with the structural findings.

Overall, this is an impressive work and greatly expands our understanding of the mechanism of RanBP2 in nuclear export and SUMO-related functions. This manuscript will make an exceptional addition to the journal. I only have a few minor points as listed below.

We thank the reviewer for their synopsis and support.

1. Fig 1A: It would be helpful to define some of the domain abbreviations (such as OE, RBD) in the legend.

We added several domain definitions that are relevant to the structure to the figure legend.

2. Comparison with PDB 1K5D (*S. pombe* RanGAP1) on pg 8 lacks context. Perhaps a supplemental figure can be added.

We can only compare our structure to the *S. pombe* PDB 1K5D structure since the latter is the only currently available structure of RanGAP1 in complex with Ran and a RBD. We understand that the comparison may seem out of context and to address that, we edited the sentence in the revised manuscript to read “This subcomplex structure is similar to the only currently available structure of RanGAP1 in complex with Ran, a crystal structure of the *S. pombe* RanGAP1 in complex with human Ran(GMPPNP) and the RBD of RanBP1 (PDB 1K5D).”

3. Catalytic glutamine of Ran (pg 8): The residue number should be pointed out in the text.

We added the residue number in the revised manuscript.

4. EMDB ID should be added in Extended Data Figures 1 and 2, and Supplementary Table 1 as applicable. Currently, it is hard to correlate the various maps with the EMDB entries.

We added the EMDB IDs in the revised Supplementary Figures 2 and 3.

We thank the reviewers for their insightful comments and suggestions and for their positive reviews of the manuscript. In response to reviewer comments, we have made the following major changes to the manuscript:

The following point-by-point responses outline our changes to the revised manuscript as per reviewers' critiques.

Reviewer #1 (Remarks to the Author):

Baytshtok and colleagues present a compelling manuscript addressing the structural and functional characterization of a challenging and conformationally dynamic complex comprising the human nuclear export factor CRM1•Ran(GTP) with the cytoplasmic filament nucleoporin constituent RanBP2 (NUP358)•SUMO1-RanGAP1•UBC9, which localizes either at the NPC or, during mitosis is recruited to the Kinetochore. Overall, the revisions implemented in response to both my feedback and that of the other reviewers has yielded a polished manuscript that markedly enhances the rigor of the experimental SEC reconstitutions and the clarity of the graphical presentations compared to the previous submission. I particularly appreciated the comprehensive reconstitution experiments with appropriate individual protein controls, presented in Figure S1; and the refined graphical layout in Figure 2A, which clarify the cryo-EM workflow and succinctly highlights regions exhibiting conformational heterogeneity. As all my comments have been addressed, I reiterate my strong support for the timely publication of this punchy and meticulously prepared manuscript in Nature Communications without further revision.

Response: We thank the reviewer for their comments and support.

Reviewer #2 (Remarks to the Author):

This revised manuscript (NCOMMS-25-06168A) addressed most of my concerns on previous manuscript. I appreciate that they provided additional data for in vitro SUMOylation assay with Leptomycin B and performed SEC reconstitution assay with additional controls. Discussion section is improved by addressing multiple comments from reviewers.

Overall, I truly appreciate the authors' efforts on addressing reviewers' concerns to strengthen the conclusion of this study. I fully support publishing this manuscript in the Nature communications.

Response: We thank the reviewer for their comments and support.

Reviewer #3 (Remarks to the Author):

The suggestions made have been fully implemented in the revised version of the manuscript and I have no further comments. I strongly recommend publication in Nature communications.

Response: We thank the reviewer for their comments and support.